# Implicit Regularization of Bregman Proximal Point Algorithm and Mirror Descent on Separable Data

## Abstract

Bregman proximal point algorithm (BPPA) has witnessed emerging machine learning applications, yet its theoretical understanding has been largely unexplored. We study the computational properties of BPPA through learning linear classifiers with separable data, and demonstrate provable algorithmic regularization of BPPA. For any BPPA instantiated with a fixed Bregman divergence, we provide a lower bound of the margin obtained by BPPA with respect to an arbitrarily chosen norm. The obtained margin lower bound differs from the maximal margin by a multiplicative factor, which inversely depends on the condition number of the distance-generating function measured in the dual norm. We show that the dependence on the condition number is tight, thus demonstrating the importance of divergence in affecting the quality of the learned classifiers. We then extend our findings to mirror descent, for which we establish similar connections between the margin and Bregman divergence, together with a non-asymptotic analysis. Numerical experiments on both synthetic and real-world datasets are provided to support our theoretical findings. To the best of our knowledge, the aforementioned findings appear to be new in the literature of algorithmic regularization.

## 1 Introduction

The role of optimization methods has become arguably one of the most critical factors in the empirical performances of machine learning models. As the go-to choice in practice, first-order algorithms, including (stochastic) gradient descent and their adaptive counterparts (Kingma and Ba, 2014; Duchi et al., 2011), have received tremendous attention, with detailed investigations dedicated to understanding the effect of various algorithmic designs, including batch size (Goyal et al., 2017; Smith et al., 2018), learning rate (Li et al., 2019; He et al., 2019; Lewkowycz et al., 2020), momentum (Sutskever et al., 2013; Smith, 2018).

Meanwhile, Bregman proximal point algorithm (BPPA) (Eckstein, 1993; Kiwiel, 1997; Yang and Toh, 2021), a classical non-first-order method that was relatively underexplored in the machine learning community, has been drawing rising interests. The successes of this classical method are particularly evident in knowledge distillation (Furlanello et al., 2018), mean-teacher learning paradigm (Tarvainen and Valpola, 2017), few-shot learning (Zhou et al., 2019), policy optimization (Green et al., 2019), and fine-tuning pre-trained models (Jiang et al., 2020), yielding competitive performance compared to its first-order counterparts.

In the general form[1], BPPA updates parameters by minimizing a loss $\mathcal{L} : \mathbb{R}^d \to \mathbb{R}$, while regularizing the weighted distance to the previous iterate measured by some divergence function $\mathcal{D} : \mathbb{R}^d \times \mathbb{R}^d \to \mathbb{R}_+$,

$$\theta_{t+1} \in \underset{\theta \in \mathbb{R}^d}{\operatorname{argmin}} \, \mathcal{L}(\theta) + \frac{1}{2\eta_t} \mathcal{D}(\theta, \theta_t), \tag{1}$$

where $\{\eta_t\}$ serves as the stepsizes of BPPA. Such a simple update is of practical purposes, as it is easy to describe and implement by adopting suitable off-the-shelf black-box optimization methods (Solodov and Svaiter, 2000; Monteiro and Svaiter, 2010; Zaslavski, 2010). The update (1) also suggests some plausible intuitions for its empirical successes, including iteratively constraining the search space, alleviating aggressive

---

[1]See Section 2 for a formal descriptions of Bregman divergence and BPPA update.

updates, and preventing catastrophic forgetting (Schulman et al., 2015; Li and Hoiem, 2017). However, none of the intuitions has been rigorously justified, and there seems to be limited effort devoted to understanding the empirical successes of BPPA, a sharp contrast when compared to its first-order counterparts.

This paper aims to provide an initiation to the currently limited theoretical understandings on the behavior of Bregman proximal point algorithm in training machine learning models. Our point of focus goes beyond the optimization properties of BPPA often well studied in the optimization literature. Instead, most of our development will be devoted to studying the structural properties of the produced solution from BPPA by drawing motivations from the following core questions.

The first and a seemingly natural question is whether Bregman proximal point algorithm benefits from similar mechanisms of (stochastic) gradient descent (GD/SGD). In particular, when the training objective has non-unique solutions (i.e., the optimization problem being under-determined), GD/SGD is widely believed to converge to solutions with favorable structural properties. Such a claim is supported with numerous provable examples: GD/SGD converges to the minimum-norm solution of under-determined linear systems (folklore, see also Gunasekar et al. (2018)), converges to the max-margin solution for separable data (Soudry et al., 2018; Nacson et al., 2019; Li et al., 2020), aligns layers of deep linear networks (Ji and Telgarsky, 2018), and converges to a generalizable solution for nonlinear networks (Brutzkus et al., 2017; Allen-Zhu et al., 2018) in the presence overfitting solutions. Given the aforementioned evidences on its first-order counterparts finding well-structured solutions, one would naturally ask

> **Q1:** Does BPPA converge to any structured solution with favorable properties?

It should be mentioned that by varying the choice of divergence $\mathcal{D}$ in (1), BPPA would generate different iterate sequences, which can lead to different computational and structural properties of the final solution. Accordingly, we will explicitly state that update (1) is BPPA instantiated by divergence $\mathcal{D}$. In view of this observation, explicit role of divergence $\mathcal{D}$ has to be taken into into account when proposing an answer to **Q1**. Being motivated in theoretical nature, this point is indeed strongly supported by numerous empirical evidences, where successful applications of BPPA hinges upon a careful design of divergence measure (Li and Hoiem, 2017; Hinton et al., 2015; Jiang et al., 2020). Identifying the underlying mechanism for the success or failure of a given divergence choice is not only of theoretical interest, but can reduce human effort in searching/designing the suitable divergence for a given task.

As an important addition, it can be natural to ask whether the impact of divergence on BPPA finds counterparts in its first-order counterpart mirror descent (MD, Nemirovski and Yudin (1983)). In such cases, better task-dependent algorithmic designs could be proposed with the use of suitable divergences.

Given our prior discussions, we raise the following refinement to **Q1**.

> **Q2:** How does divergence affect the structure of the solution obtained by BPPA (and other first-order algorithms)?

In addressing **Q1** and **Q2**, the problem we consider is a simple yet nontrivial under-determined system: training linear classifiers on separable data. In particular, for exponential-tailed losses (e.g., exponential/logistic loss), the training objective has infimum zero that is only asymptotically attainable at infinity by traversing within certain cone. The central structural property we focus on the obtained classifier is its margin, defined as the minimum distance between the samples and the decision hyperplane and is tractably computable for linear classifiers. For such a problem, we are able to provide concrete answers to the proposed questions, by establishing tight connections between divergence and the margin properties of BPPA and its first-order counterpart MD. In summary, our contributions can be categorized into the following aspects.

First, for any fixed Bregman divergence, we show that the instantiated BPPA attains a nontrivial margin lower bound, measured in an arbitrarily chosen norm. Notably no knowledge of the norm is required by the method itself. In particular, for any chosen norm, the obtained margin lower bound inversely depends on the condition number of the distance-generating function (DGF, see Definition 4) instantiating the BPPA,

where the condition number is measured with respect to the dual norm. En route, we provide a non-asymptotic analysis of the margin progress and the optimality gap of the training objective. We establish convergence rate of the two quantities for constant stepsize BPPA. We further propose a BPPA variant with more agressive stepsizes that shows exponential speedup for both quantities.

Second, we show that the dependence of the margin lower bound on the condition number of DGF established before is indeed tight. Specifically, we construct a class of problems of increasing dimensions, where the limiting solution of the instantiated BPPA applied to each problem attains a margin that is at most twice of the lower bound. In addition, the obtained margin diminishes to zero as dimension increases to infinity, thus illustrating the importance of using the correct divergence (DGF) when instantiating BPPA.

Third, we extend our findings to first-order methods. Specifically, we show that mirror descent (MD) exhibits similar connections between the margin and the divergence. We also provide non-asymptotic convergence analyses of the margin and optimality gap for constant stepsize MD, and establish an exponential speed-up using an adaptive stepsize scheme. Our findings for MD seem to strictly complement prior works with exclusive focuses on under-determined regression problems (Gunasekar et al., 2018; Azizan and Hassibi, 2019; Wu and Rebeschini, 2021), while focusing on more challenging classification tasks.

Finally, we conduct numerical experiments on both synthetic and real datasets using linear models and nonlinear neural networks. Our experiments verify the theoretical development, and demonstrate that the obtained results for linear models can potentially carry over to training more complex models.

The rest of the paper is organized as follows. Section 2 introduces the problem setup and the BPPA method. Section 3 presents our main theoretical findings of BPPA. Section 4 extends our findings to mirror descent method. Section 5 presents numerical study to support our developed theories. Concluding remarks are made in Section 6.

**Notations**. We denote $[n] \coloneqq \{1, \ldots, n\}$; $\mathrm{sgn}(z) = 1$ if $z \geq 0$ and $-1$ elsewhere. We use w.r.t in short for "with respect to". For any $\|\cdot\|$ in Euclidean space $\mathbb{R}^d$, we use $\|\cdot\|_* = \max_{\|y\| \leq 1} \langle \cdot, y \rangle$ to denote its dual norm.

## 2   Problem Setup

Consider a binary classification task, where $\mathcal{S} = \{(x_i, y_i)\}_{i=1}^n \subset \mathbb{R}^d \times \{+1, -1\}$ denotes the dataset, $x_i$ denotes the feature vector, and $y_i$ denotes the label. The data is linearly separable in the sense that there exists a linear classifier $u \in \mathbb{R}^d$, such that $y_i \langle u, x_i \rangle > 0$ for all $i \in [n]$. That is, the decision rule $f_u(\cdot) = \mathrm{sgn}(\langle u, \cdot \rangle)$ attains perfect accuracy on the dataset, with $y_i = f_u(x_i)$ for all $i \in [n]$. For each linear classifier $f_u$ with perfect accuracy and any norm $\|\cdot\|$, we define the dual norm margin of $f_u$ as follows.

**Definition 2.1** ($\|\cdot\|_*$-norm Margin). *For each linear classifier $f_u$ with perfect accuracy and any norm $\|\cdot\|$, the $\|\cdot\|_*$-norm margin of $f_u$ is defined as*

$$\gamma_{u, \|\cdot\|_*} \coloneqq \min_{i \in [n]} \left\langle x_i y_i, \frac{u}{\|u\|} \right\rangle.$$

*That is, $\gamma_{u, \|\cdot\|_*}$ is the minimum distance from the feature vectors to the decision boundary $\mathcal{H}_u = \{x : \langle x, u \rangle = 0\}$ measured in $\|\cdot\|_*$-norm.*

The $\|\cdot\|_*$-norm margin measures how well the data is separated by decision rule $f_u$, measured in $\|\cdot\|_*$-norm, and is an important measure on the generalizability and robustness of the decision rule (Koltchinskii and Panchenko, 2002; Bartlett and Mendelson, 2002; Xu and Mannor, 2012). The optimal linear classifier with the maximum $\|\cdot\|_*$-margin is defined as follows.

**Definition 2.2** (Maximum $\|\cdot\|_*$-norm Margin Classifier). *Given a linearly separable dataset $\{(x_i, y_i)\}_{i \in [n]}$, we define the maximum $\|\cdot\|_*$-norm margin classifier $u_{\|\cdot\|_*}$, and its associated maximum $\|\cdot\|_*$-norm margin $\gamma_{\|\cdot\|_*}$ as*

$$u_{\|\cdot\|_*} \coloneqq \operatorname*{argmax}_{\|u\| \leq 1} \min_{i \in [n]} \langle u, y_i x_i \rangle, \quad \gamma_{\|\cdot\|_*} \coloneqq \max_{\|u\| \leq 1} \min_{i \in [n]} \langle u, y_i x_i \rangle.$$

---

**Algorithm 1** BPPA Instantiated by DGF $w(\cdot)$

---

**Input:** Distance-generating function $w : \mathbb{R}^d \to \mathbb{R}$, stepsizes $\{\eta^t\}_{t \geq 0}$, samples $\{x_i, y_i\}_{i=1}^n$.
**Initialize:** $\theta^0 \leftarrow 0$.
**for** $t = 0, \dots$ **do**

$$\text{Update: } \theta_{t+1} \in \operatorname{argmin}_{\theta \in \mathbb{R}^d} \mathcal{L}(\theta) + \frac{1}{2\eta_t} D_w(\theta, \theta_t). \tag{2}$$

**end for**

---

We consider the problem of learning a linear classifier by minimizing the empirical loss

$$\mathcal{L}(\theta) = \frac{1}{n} \sum_{i=1}^n \ell\left(\langle \theta, y_i x_i \rangle\right). \tag{3}$$

Going forward we focus on the exponential loss $\ell(x) = \exp(-x)$ for presentation simplicity, while the analyses can be readily extended to other losses with tight exponential tail (e.g., logistic loss).

It is worth mentioning that with a separable dataset $\mathcal{S}$, the empirical loss has infimum 0 but possesses no finite solution attaining the infimum. To see this, note that $\mathcal{L}(\lambda u_{\|\cdot\|_*}) \to 0$ as $\lambda \to 0$, and hence $\inf_{\theta \in \mathbb{R}^d} \mathcal{L}(\theta) = 0$, while $\mathcal{L}(\theta) > 0$ for any $\theta$. As a consequence, any iterative method that minimizes (3) will observe $\|\theta_k\| \to \infty$ as $\mathcal{L}(\theta_k) \to 0$. Moreover, it should also be clear that $u_{\|\cdot\|_*}$ is not the only direction for which we can drive the loss to zero. In particular, define $\mathcal{Z} = \{y_i x_i : i \in [n]\}$, and let $\mathcal{Z}^* = \{u : u^\top z > 0, \forall z \in \mathcal{Z}\}$. It is clear that any direction in cone $\mathcal{Z}^*$ is a valid direction. As $u_{\|\cdot\|_*} \in \mathcal{Z}^*$, one can readily verify that $\mathcal{Z}^*$ has a nonempty interior.

The Bregman Proximal Point Algorithm (BPPA, Algorithm 1) (Eckstein, 1993; Kiwiel, 1997; Yang and Toh, 2021) is an important extension of the vanilla proximal point method (Rockafellar, 1976a;b) to non-euclidean geometry. In particular, each step of the method makes use of the so-called Bregman divergence, defined as follows.

**Definition 2.3** (Bregman Divergence). *Given $w : \mathbb{R}^d \to \mathbb{R}$ that is convex and differentiable, we define the Bregman divergence $D_w$ associated with $w$ as*

$$D_w(\theta, \theta') = w(\theta) - w(\theta') - \langle \nabla w(\theta'), \theta - \theta' \rangle, \tag{4}$$

*and refer to $w$ as the distance-generating function of the Bregman divergence.*

Each step of BPPA applied to problem (3) takes the form of

$$\theta_{t+1} \in \operatorname*{argmin}_{\theta \in \mathbb{R}^d} \mathcal{L}(\theta) + \frac{1}{2\eta_t} D_w(\theta, \theta_t), \tag{5}$$

where $\{\eta_t\}$ denotes the stepsizes. Since different $w(\cdot)$ leads to different iterates $\{\theta_t\}$, we will explicitly refer to update (5) as BPPA instantiated by distance-generating function $w(\cdot)$. In particular, when $w(\theta) = \|\theta\|_2^2$, BPPA instantiated by $w(\cdot)$ recovers the vanilla proximal point method.

We make the following sole condition on the distance-generating function for the remainder of our discussions.

**Condition 1.** *The distance generating function $w(\cdot)$ is $L_{\|\cdot\|}$-smooth and $\mu_{\|\cdot\|}$-strongly convex w.r.t. $\|\cdot\|$. That is,*

$$\frac{\mu_{\|\cdot\|}}{2} \|\theta - \theta'\|^2 \leq w(\theta) - w(\theta') - \langle \nabla w(\theta'), \theta - \theta' \rangle \leq \frac{L_{\|\cdot\|}}{2} \|\theta - \theta'\|^2, \quad \forall \theta, \theta' \in \mathbb{R}^d. \tag{6}$$

## 3 Algorithmic Regularization of Bregman Proximal Point Algorithm

We start by discussing the convergence of BPPA applied to (3) and establish the margin lower bound for the limiting solution.

**Theorem 3.1** (Constant Stepsize BPPA). *Let $D_{\|\cdot\|_*} = \max_{i \in [n]} \|x_i\|_*$. Then under Condition 1, for any constant stepsize $\eta_t = \eta > 0$, the following holds.*

*(1) We have*

$$\mathcal{L}(\theta_t) \leq \frac{1}{\gamma_{\|\cdot\|_*}\eta t} + \frac{L_{\|\cdot\|}\log^2\left(\gamma_{\|\cdot\|_*}\eta t\right)}{4\gamma_{\|\cdot\|_*}^2\eta t} = \mathcal{O}\left(\frac{L_{\|\cdot\|}\log^2\left(\gamma_{\|\cdot\|_*}\eta t\right)}{\gamma_{\|\cdot\|_*}^2\eta t}\right).$$

*(2) The margin is asymptotically lower bounded by*

$$\lim_{t\to\infty}\min_{i\in[n]}\left\langle\frac{\theta_t}{\|\theta_t\|}, y_i x_i\right\rangle \geq \sqrt{\frac{\mu_{\|\cdot\|}}{L_{\|\cdot\|}}}\gamma_{\|\cdot\|_*}, \tag{7}$$

*where $\gamma_{\|\cdot\|_*}$ is defined in Definition 2.2. In addition, for any given $\epsilon > 0$, there exists*

$$t_0 = \widetilde{\mathcal{O}}\left(\max\left\{\frac{D_{\|\cdot\|_*}^2}{\epsilon^2\gamma_{\|\cdot\|_*}^2}, \exp\left(\frac{D_{\|\cdot\|_*}^2}{\gamma_{\|\cdot\|_*}^2\epsilon^2}\sqrt{\frac{L_{\|\cdot\|}}{\mu_{\|\cdot\|}}}\right)\frac{1}{\gamma_{\|\cdot\|_*}^2\eta}\right\}\right),$$

*such that for $t \geq t_0$ number of iterations, we have*

$$\left\langle\frac{\theta_t}{\|\theta_t\|}, y_i x_i\right\rangle \geq (1-\epsilon)\sqrt{\frac{\mu_{\|\cdot\|}}{L_{\|\cdot\|}}}\gamma_{\|\cdot\|_*}, \quad \forall i \in [n].$$

Before we proceed, a few remarks are in order for interpreting results in Theorem 3.1: First, the choice of Bregman divergence in BPPA is flexible and can be *data dependent*. Properly chosen data-dependent divergence can adapt to data geometry better than data-independent divergence, leading to better separation and margin. In Section 5 we demonstrate how BPPA can benefit significantly from such an adaptivity of carefully designed data-dependent divergence. Second, the convergence analysis requires handling non-finite minimizers, and the optimization problem of our interest does not meet the standard assumptions in the classical analysis of BPPA. Third, BPPA is related but nevertheless should not be confused with the homotopy method in (Rosset et al., 2004), which can be viewed as tracing the one-step BPPA with progressively increasing stepsizes.

It might be worth stressing here that working with Bregman divergence poses unique challenges, as it becomes much harder to track the iterates in the primal space (i.e., model parameters). It is known that the update of dual variables in BPPA or mirror descent has an interpretation closer to the standard gradient descent (Beck, 2017), hence making the dual space more amenable for analysis. However, to the best of knowledge, it seems previously unclear how to relate the primal margin progress to the trajectory of dual variables. Our analyses directly tackle these challenges, which we view as our main technical contributions in this work.

Theorem 3.1 shows that if the distance generating function $w(\cdot)$ is well-conditioned w.r.t. $\|\cdot\|$-norm, then Bregman proximal point algorithm outputs a solution with near optimal $\|\cdot\|_*$-norm margin. As a concrete realization of Theorem 3.1, we consider the Mahalanobis distance $\|\cdot\|_A \coloneqq \sqrt{\langle\cdot, A\cdot\rangle}$ induced by a positive definite matrix $A$. It can be noted that for any positive definite $A$, the distance generating function $w(\cdot) = \langle\cdot, A\cdot\rangle$ is in fact 2-strongly convex and 2-smooth w.r.t. norm $\|\cdot\|_A$. Thus applying Theorem 3.1, we have the following corollary.

**Corollary 3.1.** *Let $\|\cdot\| = \|\cdot\|_A$ for some positive definite matrix $A$. Under the same conditions as in Theorem 3.1, BPPA with distance generating function $w(\cdot) = \langle\cdot, A\cdot\rangle$ converges to the maximum $\|\cdot\|_*$-margin solution, where $\|\cdot\|_* = \|\cdot\|_{A^{-1}}$. Specifically, we have*

$$\mathcal{L}(\theta_t) \leq \frac{1}{\gamma_{\|\cdot\|_*}\eta t} + \frac{L_{\|\cdot\|}\log^2\left(\gamma_{\|\cdot\|_*}\eta t\right)}{4\gamma_{\|\cdot\|_*}^2\eta t} = \mathcal{O}\left(\frac{L_{\|\cdot\|}\log^2\left(\gamma_{\|\cdot\|_*}\eta t\right)}{\gamma_{\|\cdot\|_*}^2\eta t}\right).$$

*In addition, $\lim_{t\to\infty}\min_{i\in[n]}\left\langle\frac{\theta_t}{\|\theta_t\|}, y_i x_i\right\rangle = \gamma_{\|\cdot\|_*}$. For any given $\epsilon > 0$, there exists*

$$t_0 = \widetilde{\mathcal{O}}\left(\max\left\{\frac{D_{\|\cdot\|_*}^2}{\epsilon^2\gamma_{\|\cdot\|_*}^2}, \exp\left(\frac{D_{\|\cdot\|_*}^2}{\gamma_{\|\cdot\|_*}^2\epsilon^2}\right)\frac{1}{\gamma_{\|\cdot\|_*}^2\eta}\right\}\right),$$

*such that for $t \geq t_0$ number of iterations, it holds that*

$$\left\langle \frac{\theta_t}{\|\theta_t\|}, y_i x_i \right\rangle \geq (1 - \epsilon)\gamma_{\|\cdot\|_*}, \quad \forall i \in [n].$$

*Finally, we have the directional convergence such that $\lim_{t \to \infty} \frac{\theta_t}{\|\theta_t\|} = u_{\|\cdot\|_*}$.*

When the distance generating function $w(\cdot)$ is ill-conditioned w.r.t. $\|\cdot\|$-norm (i.e., $\sqrt{\mu_{\|\cdot\|}/L_{\|\cdot\|}} \ll 1$), it might be tempting to suggest that the margin lower bound in (7) is loose. However, as we show in the following proposition, there exists a class of problems where the lower bound in (7) is in fact a tight upper bound (up to a factor of 2), demonstrating that the dependence on condition number of distance generating function $w(\cdot)$ is essential.

**Theorem 3.2** (Tight Dependence on Condition Number)**.** *There exists a sequence of problems $\{\mathcal{P}^{(m)}\}_{m \geq 1}$, where each $\mathcal{P}^{(m)} = \left( \mathcal{S}^{(m)}, \|\cdot\|^{(m)}, w^{(m)} \right)$ denotes the dataset, the norm, and the distance generating function of the $m$-th problem. For each $m$, the distance generating function $w^{(m)}(\cdot)$ is $\mu_{\|\cdot\|}^{(m)}$-strongly convex and $L_{\|\cdot\|}^{(m)}$-smooth w.r.t. $\|\cdot\|$-norm. The Bregman proximal point algorithm applied to each problem in $\{\mathcal{P}^{(m)}\}_{m \geq 1}$ yields*

$$\lim_{t \to \infty} \min_{(x,y) \in \mathcal{S}^{(m)}} \left\langle \frac{\theta_t}{\|\theta_t\|}, yx \right\rangle \Big/ \gamma_{\|\cdot\|_*^{(m)}} \leq 2\sqrt{\frac{\mu_{\|\cdot\|}^{(m)}}{L_{\|\cdot\|}^{(m)}}}, \quad \forall m \geq 1, \tag{8}$$

*In addition, for any $m \geq 4$, we have $\lim_{t \to \infty} \min_{(x,y) \in \mathcal{S}^{(m)}} \left\langle \frac{\theta_t}{\|\theta_t\|}, yx \right\rangle \Big/ \gamma_{\|\cdot\|_*^{(m)}} \leq 2\sqrt{\frac{\mu_{\|\cdot\|}^{(m)}}{L_{\|\cdot\|}^{(m)}}} < 1$. In fact,*

$$\lim_{t \to \infty} \min_{(x,y) \in \mathcal{S}^{(m)}} \left\langle \frac{\theta_t}{\|\theta_t\|}, yx \right\rangle \Big/ \gamma_{\|\cdot\|_*^{(m)}} \to 0, \quad as \ m \to \infty. \tag{9}$$

*Remark* 3.1. Combining Theorem 3.1, Corollary 3.1 and Theorem 3.2, it can be seen that the condition number of the distance generating function has an essential role in determining the margin of the obtained solution by BPPA. This observation advocates a careful design of Bregman divergence in search for well-structured solutions. The findings also align with the empirical evidences on the importance of divergence found in applications of knowledge distillation and model fine-tuning (Jiang et al., 2020; Furlanello et al., 2018).

We have shown that BPPA with constant stepsize achieves a margin that is at least $\sqrt{\mu_{\|\cdot\|}/L_{\|\cdot\|}}$-fraction of the maximal one. Meanwhile, Theorem 3.1 indicates that to obtain such a margin lower bound, it might take an exponential number of iterations. We proceed to establish that by employing a more aggressive stepsize scheme, one can attain the same margin lower bound in a polynomial number of iterations and speed up the convergence of the empirical loss drastically.

**Theorem 3.3** (Varying Stepsize BPPA)**.** *Given any positive sequence $\{\alpha_t\}_{t \geq 0}$, letting the stepsizes $\{\eta_t\}_{t \geq 0}$ be $\eta_t = \frac{\alpha_t}{\mathcal{L}(\theta_t)}$, then the following facts hold.*

*(1) $\lim_{t \to \infty} \mathcal{L}(\theta_t) = 0$. Specifically, for any $t \geq 0$, we have $\mathcal{L}(\theta_{t+1}) \leq \mathcal{L}(\theta_t)\beta(\alpha_t)$, where $\beta(\alpha) = \min_{\beta \in (0,1)} \max \left\{ \beta, \exp\left( -\frac{2\alpha\beta^2\gamma_{\|\cdot\|_*}^2}{L_{\|\cdot\|}} \right) \right\} < 1$.*

*(2) Letting $\alpha_t = \frac{1}{\sqrt{t+1}}$, we have $\lim_{t \to \infty} \min_{i \in [n]} \left\langle \frac{\theta_t}{\|\theta_t\|}, y_i x_i \right\rangle \geq \sqrt{\frac{\mu_{\|\cdot\|}}{L_{\|\cdot\|}}}\gamma_{\|\cdot\|_*}$. In particular, for any $\epsilon \in (0, \frac{1}{2})$, there exists a $t_0$ satisfying*

$$t_0 = \mathcal{O}\left( \left( \frac{L_{\|\cdot\|}}{\gamma_{\|\cdot\|_*}\sqrt{\mu_{\|\cdot\|}}\epsilon} \right)^8 \right), \tag{10}$$

*such that in $t \geq t_0$ number of iterations, we have*

$$\left\langle \frac{\theta_t}{\|\theta_t\|}, y_i x_i \right\rangle \geq (1-\epsilon) \sqrt{\frac{L_{\|\cdot\|}}{\mu_{\|\cdot\|}}} \gamma_{\|\cdot\|_*}, \quad \forall i \in [n].$$

*Additionally, the empirical loss diminishes at the following rate:*

$$\mathcal{L}(\theta_t) = \mathcal{O}\left( \exp\left( -\frac{\gamma_{\|\cdot\|_*}^2}{L_{\|\cdot\|}} \sqrt{t} \right) \right).$$

A few remarks are in order for interpreting Theorem 3.3. First, we do not optimize for the best polynomial dependence on $1/\epsilon$ in the iteration complexity (10), as our goal is to illustrate the exponential gap between the complexity presented in Theorem 3.1 and here. We refer interested readers to Appendix C, where we show an improved polynomial dependence can be potentially obtained with a more refined analysis. It is also worth noting that using the aggressive stepsizes does not change our established margin lower bound, and the exact convergence to the maximum margin solution demonstrated in Corollary 3.1 still holds for this stepsize scheme.

● **Inexact Implementation of BPPA.** The proximal update (2) requires solving a non-trivial optimization problem, and there has been fruitful results of inexact implementation of BPPA in optimization literature (Rockafellar, 1976b; Yang and Toh, 2021; Solodov and Svaiter, 2000; Monteiro and Svaiter, 2010). Here based on the varying stepsize scheme proposed in Theorem 3.3, we discuss the feasibility of a gradient descent based inexact BPPA that admits a simple implementation and achieves polynomial complexity, while retaining the margin properties of exact BPPA. Specifically, at the $t$-th iteration, the gradient descent based inexact BPPA solves the proximal step

$$\widehat{\theta}_{t+1} \approx \underset{\theta}{\operatorname{argmin}}\, \phi_t(\theta) := \frac{1}{n} \sum_{i=1}^n \exp\left( -\langle \theta, y_i x_i \rangle \right) + \frac{1}{2\eta_t} D_w(\theta, \widehat{\theta}_t) \tag{11}$$

up to a pre-specified accuracy $\delta_t$ with gradient descent. It can be noted that when applying gradient descent to $\phi_t(\cdot)$ with small enough stepsizes, the iterate would stay in a region that has relative smoothness $M_t$ and relative strong convexity $\mu_t$ bounded by

$$M_t \leq \mathcal{L}(\widehat{\theta}_t) + \frac{1}{\eta_t} = \mathcal{L}(\widehat{\theta}_t)\left( 1 + \frac{1}{\alpha_t} \right), \quad \mu_t \geq \frac{1}{\eta_t} = \frac{L(\widehat{\theta}_t)}{\alpha_t},$$

both measured w.r.t. Bregman divergence $D_h(\cdot, \cdot)$ (Lu et al., 2018). Note that the first inequality follows by our choice of stepsize $\eta_t$ in Theorem 3.3. Thus the effective condition number $\kappa_t := M_t/\mu_t$ of $\phi_t(\cdot)$ is bounded by $\kappa_t = 1 + \alpha_t = \mathcal{O}(1)$, which implies that the $t$-th proximal step requires $\mathcal{O}\left( \kappa_t \log(\frac{1}{\delta_t}) \right) = \mathcal{O}\left( \log(\frac{1}{\delta_t}) \right)$ number of gradient descent steps. Summing up across $t_0$ iterations defined in (10), it suffices to take $\mathcal{O}\left( \sum_{t=1}^{t_0} \log(\frac{1}{\delta_t}) \right)$ gradient descent steps, which depends polynomially on $t_0$ even if we choose high accuracy $\delta_t = \mathcal{O}(\exp(-t))$ for performing each inexact proximal step considered in (11).

## 4 Algorithmic Regularization of Mirror Descent

We proceed to establish that mirror descent (MD, Algorithm 2), as a generalization of gradient descent to non-euclidean geometry, possesses similar connections between the margin and Bregman divergence. It might be worth noting here that the to-be-developed results are the first to characterize the algorithmic regularization effect of MD for classification tasks, while previous studies focus on under-determined regression problems (Gunasekar et al., 2018; Azizan and Hassibi, 2019).

**Theorem 4.1** (Constant Stepsize MD). *Let $D_{\|\cdot\|_*} = \max_{i \in [n]} \|x_i\|_*$, where $\|\cdot\|_*$ denotes the dual norm of $\|\cdot\|$, and $D_{\|\cdot\|_2} = \max_{i \in [n]} \|x_i\|_2$. Let $\mu_2$ be the strong convexity modulus of $w(\cdot)$ w.r.t. $\|\cdot\|_2$-norm[2]. Then for any constant stepsize $\eta_t = \eta \leq \frac{\mu_2 \mu_{\|\cdot\|}}{2 L_{\|\cdot\|} D_{\|\cdot\|_2}}$, we have the following facts.*

---

[2]It can be seen that $\mu_2 > 0$ under Condition 1 and the equivalence of norm in $\mathbb{R}^d$.

---

**Algorithm 2** Mirror Descent Algorithm (MD)

---

**Input:** Distance generating function $w(\cdot)$, stepsizes $\{\eta^t\}_{t \geq 0}$, samples $\{x_i, y_i\}_{i=1}^n$.
**Initialize:** $\theta^0 \leftarrow 0$.
**for** $t = 0, \dots$ **do**
    Compute gradient $\nabla \mathcal{L}(\theta_t) = \frac{1}{n} \sum_{i=1}^n \exp\left(-\langle \theta_t, y_i x_i \rangle\right)(-y_i x_i)$.
    Update $\theta_{t+1} = \arg\min_\theta \langle \nabla \mathcal{L}(\theta_t), \theta - \theta_t \rangle + \frac{1}{2\eta_t} D_w(\theta, \theta_t)$.
**end for**

---

*(1)* $\lim_{t \to \infty} \mathcal{L}(\theta_t) = 0$. *Specifically,* $\mathcal{L}(\theta_t)$ *diminishes at the following rate:*

$$\mathcal{L}(\theta_t) \leq \frac{1}{\gamma_{\|\cdot\|_*} \eta t} + \frac{L_{\|\cdot\|} \log^2\left(\gamma_{\|\cdot\|_*} \eta t\right)}{4\gamma_{\|\cdot\|_*}^2 \eta t} = \mathcal{O}\left(\frac{L_{\|\cdot\|} \log^2\left(\gamma_{\|\cdot\|_*} \eta t\right)}{\gamma_{\|\cdot\|_*}^2 \eta t}\right).$$

*(2) The margin is asymptotically lower bounded by*

$$\lim_{t \to \infty} \min_{i \in [n]} \left\langle \frac{\theta_t}{\|\theta_t\|}, y_i x_i \right\rangle \geq \sqrt{\frac{\mu_{\|\cdot\|}}{L_{\|\cdot\|}}} \gamma_{\|\cdot\|_*}.$$

*In addition, for any $\epsilon > 0$, there exists*

$$t_0 = \mathcal{O}\left(\exp\left(\frac{D_{\|\cdot\|_*}^{3/2} D_{\|\cdot\|_2} L_{\|\cdot\|} \eta}{\gamma_{\|\cdot\|_*}^2 \mu_{\|\cdot\|}^{1/2} \mu_2^{3/2} \epsilon^{3/2}} \log\left(\frac{1}{\epsilon}\right)\right)\right), \tag{12}$$

*such that any $t \geq t_0$, we have*

$$\left\langle \frac{\theta_t}{\|\theta_t\|}, y_i x_i \right\rangle \geq (1-\epsilon)\sqrt{\frac{\mu_{\|\cdot\|}}{L_{\|\cdot\|}}} \gamma_{\|\cdot\|_*}, \quad \forall i \in [n].$$

Theorem 4.1 shows that mirror descent attains the same $\|\cdot\|_*$-norm margin lower bound as BPPA, which is $\sqrt{\mu_{\|\cdot\|}/L_{\|\cdot\|}}$-fraction of the maximal margin. Similar to Corollary 3.1, let $\|\cdot\| = \|\cdot\|_A$ be the Mahalanobis distance, then MD equipped with distance generating function $w(\cdot) = \langle \cdot, A \cdot \rangle$ converges to the maximum $\|\cdot\|_*$-norm margin classifier. Note that for such a setting, MD is equivalent to the steepest descent algorithm, which has also been shown to converge to the maximum $\|\cdot\|_*$ margin (Gunasekar et al., 2018).

**Corollary 4.1.** *Let $\|\cdot\| = \|\cdot\|_A$ for some positive definite matrix $A$. Then under the same conditions as in Theorem 4.1, MD with distance generating function $w(\cdot) = \langle \cdot, A \cdot \rangle$ converges to the maximum $\|\cdot\|_*$-margin solution, where $\|\cdot\|_* = \|\cdot\|_{A^{-1}}$. Specifically, we have*

$$\mathcal{L}(\theta_t) = \mathcal{O}\left(\frac{L_{\|\cdot\|} \log^2\left(\gamma_{\|\cdot\|_*} \eta t\right)}{\gamma_{\|\cdot\|_*}^2 \eta t}\right).$$

*In addition,* $\lim_{t \to \infty} \min_{i \in [n]} \left\langle \frac{\theta_t}{\|\theta_t\|}, y_i x_i \right\rangle = \gamma_{\|\cdot\|_*}$. *For any given $\epsilon > 0$, there exists*

$$t_0 = \mathcal{O}\left(\exp\left(\frac{D_{\|\cdot\|_*}^{3/2} D_{\|\cdot\|_2} L_{\|\cdot\|} \eta}{\gamma_{\|\cdot\|_*}^2 \mu_{\|\cdot\|}^{1/2} \mu_2^{3/2} \epsilon^{3/2}} \log\left(\frac{1}{\epsilon}\right)\right)\right),$$

*such that for $t \geq t_0$, we have*

$$\left\langle \frac{\theta_t}{\|\theta_t\|}, y_i x_i \right\rangle \geq (1-\epsilon)\gamma_{\|\cdot\|_*}, \quad \forall i \in [n].$$

*Finally, we have directional convergence such that* $\lim_{t \to \infty} \frac{\theta_t}{\|\theta_t\|} = u_{\|\cdot\|_*}$.

Theorem 4.1 guarantees a near optimal $\|\cdot\|_*$-norm margin when the distance generating function $w(\cdot)$ is well-conditioned w.r.t. $\|\cdot\|$-norm. For cases when $w(\cdot)$ is ill-conditioned, we proceed to show that similar to BPPA, there exists a class of problem for which the margin lower bound obtained by MD is tight.

**Proposition 4.1.** *There exists a sequence of problems $\{\mathcal{P}^{(m)}\}_{m \geq 1}$ by the same construction as in Proposition 3.2, such that the margin lower bound in Theorem 4.1 is tight up to a non-trivial factor of 2. Specifically, we have (8) and (9) also hold for MD.*

Finally, we propose a more aggressive stepsize scheme for MD that attains the same margin lower bound. Instead of an exponential number of iterations (c.f., (12)) required by constant stepsize MD for attaining the margin lower bound, such a stepsize scheme only needs a polynomial number of iterations, and achieves an almost exponential speedup for the empirical loss.

**Theorem 4.2** (Varying Stepsize MD)**.** *Let the stepsizes $\{\eta_t\}_{t \geq 0}$ be given by $\eta_t = \frac{\alpha_t}{\mathcal{L}(\theta_t)}$, where $\alpha_t = \min\{\frac{\mu_2 \mu_{\|\cdot\|}}{2 L_{\|\cdot\|} D_{\|\cdot\|_2}}, \frac{1}{\sqrt{t+1}}\}$. Then under the same conditions as in Theorem 4.1, the following facts hold.*

*(1) We have $\lim_{t \to \infty} \min_{i \in [n]} \left\langle \frac{\theta_t}{\|\theta_t\|}, y_i x_i \right\rangle \geq \sqrt{\frac{\mu_{\|\cdot\|}}{L_{\|\cdot\|}}} \gamma_{\|\cdot\|_*}$. In particular, for any $\epsilon > 0$, there exists*

$$t_0 = \mathcal{O}\left(\left(\frac{D_{\|\cdot\|_2} L_{\|\cdot\|}}{\gamma_{\|\cdot\|_*} \mu_2 \sqrt{\mu_{\|\cdot\|}} \epsilon}\right)^4\right),$$

*such that for any $t \geq t_0$,*

$$\left\langle \frac{\theta_t}{\|\theta_t\|}, y_i x_i \right\rangle \geq (1 - \epsilon) \sqrt{\frac{\mu_{\|\cdot\|}}{L_{\|\cdot\|}}} \gamma_{\|\cdot\|_*}, \quad \forall i \in [n].$$

*(2) The empirical loss diminishes at the following rate:*

$$\mathcal{L}(\theta_t) \leq \mathcal{O}\left(\exp\left(-\frac{\gamma_{\|\cdot\|_*}^2}{L_{\|\cdot\|}} \sqrt{t}\right)\right).$$

# 5 Numerical Study

This section presents some numerical studies that verify our established results for linear models, and illustrate the potential of our prior discussion carrying over to complex nonlinear models.

## 5.1 Synthetic Data

We take $\mathcal{S} = \{((-0.5, 1), +1), ((-0.5, -1), -1), ((-0.75, -1), -1), ((2, 1), +1)\}$. One can readily verify that the maximum $\|\cdot\|_2$-norm margin classifier is $u_{\|\cdot\|_2} = (0, 1)$. For both BPPA and MD, we take the Bregman divergence as $D_w(x, y) = \|x - y\|_2^2$, which corresponds to the vanilla proximal point algorithm and gradient descent algorithm. Note that both algorithms are guaranteed to converge in direction towards $u_{\|\cdot\|_2} = (0, 1)$, following Corollary 3.1 and 4.1.

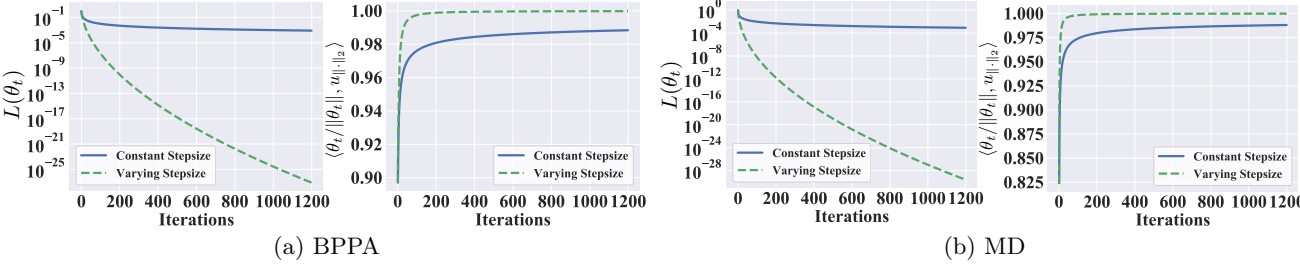

Figure 1: BPPA and MD run on the simple data set $\mathcal{S}$.

We take $\eta_t = \eta = 1$ for the constant stepsize BPPA/MD, and $\eta_t = \frac{1}{L(\theta_t)\sqrt{t+1}}$ for the varying stepsize BPPA/MD, following the stepsize choices in Theorem 3.1, 3.3, 4.1 and 4.2. To implement the proximal step in BPPA at the $t$-th iteration, we take 128 number of gradient descent steps with stepsize $0.2\eta_t$, following our discussion at the end of Section 3. We initialize all algorithms at the origin and run 1200 iterations. From Figure 1, we can clearly observe that both BPPA and MD converge in direction to the maximum $\|\cdot\|_2$-norm margin classifier $u_{\|\cdot\|_2}$, which is consistent with our theoretical findings. In addition, by adopting the varying stepsize scheme proposed in Theorem 3.3 and 4.2, both BPPA and MD converge exponentially faster than their constant stepsize counterparts.

## 5.2 Data-dependent Bregman Divergence

We illustrate through an example on how properly chosen data-dependent divergence can lead to much improved separation compared to data-independent divergence even on simple linear models.

Consider $n$ labeled data $\{(x_i, y_i)\}_{i=1}^n$ sampled from a mixture of sphere distribution: $y_i \sim \text{Bernoulli}(1/2)$, $x_i \sim \text{Unif}\,(\mathbb{S}_{y_i\mu}(r))$, where $\mathbb{S}_z(r)$ denotes the sphere centered at $z$ with radius $r$ in $\mathbb{R}^d$. In addition, we also have $m$ unlabeled data $\{\widetilde{x}_j\}_{j=1}^m$, following the same distribution as $\{x_i\}_{i=1}^n$, with no labels given. Clearly, the maximum $\|\cdot\|_2$-margin classifier for the mixture of sphere distribution considered here is given by the linear classifier $f^*(\cdot) = \text{sign}(\langle\cdot, \mu\rangle)$.

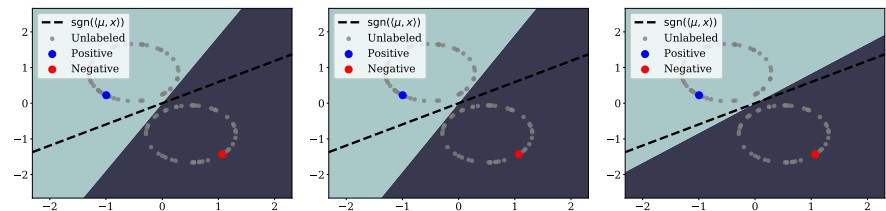

| Divergence | Alignment |
|---|---|
| $D^{(1)}(\cdot, \cdot)$ | 0.8703 |
| $D^{(2)}(\cdot, \cdot)$ | 0.8175 |
| $D^{(3)}(\cdot, \cdot)$ | 0.9754 |

Figure 2: BPPA with Bregman divergence $D^{(3)}$ (right) significantly improves alignment with optimal classifier $\mu$, compared to $D^{(1)}$ (left) and $D^{(2)}$ (middle).

Table 1: $\left\langle \frac{\theta^T}{\|\theta^T\|_2}, \mu \right\rangle$ averaged over 8 runs.

We choose $n = d = 2, m = 100, r = 0.8$, and generate $\mu \sim \text{Unif}\,(\mathbb{S}_0(1))$. We compare three types of Bregman divergence, given by $D^{(1)}(\theta, \theta') = \|\theta - \theta'\|_2^2$ (vanilla proximal point), $D^{(2)}(\theta, \theta') = (\theta - \theta')^\top \widehat{\Sigma}(\theta - \theta')$, and $D^{(3)}(\theta, \theta') = (\theta - \theta')^\top \widehat{\Sigma}^{-1}(\theta - \theta')$, where $\widehat{\Sigma} = \frac{1}{m}\sum_{j=1}^m \widetilde{x}_j \widetilde{x}_j^\top$ denotes the empirical covariance matrix. Note that $D^{(2)}$ and $D^{(3)}$ are data-dependent from their construction. For each divergence function, we run BPPA with 8 independent runs, the results are reported in Figure 2 and Table 1. We make two important remarks on the empirical results:

- Data-dependent divergence $D^{(3)}$ gives the best separation despite limited labeled data (in fact only 2!), much improved over data-independent squared $\ell_2$-distance $D^{(1)}$.

- Not all data-dependent divergence helps, $D^{(2)}$ shows degradation compared to $D^{(1)}$.

We further remark that by utilizing Corollary 3.1, one can completely characterize the solution obtained by BPPA for each of the divergence in closed form. Using such a characterization allows one to corroborate the empirical phenomenon with our developed theories, deferred in Appendix A.

## 5.3 CIFAR-100

We demonstrate the potential of extending our theoretical findings for linear models to practical networks, using ResNet-18 (He et al., 2016), ShuffleNetV2 (Ma et al., 2018), MobileNetV2 (Sandler et al., 2018), with CIFAR-100 dataset (Krizhevsky et al., 2009). At each iteration of BPPA, the updated model parameter $\theta_{t+1}$ is given by solving the proximal step

$$\theta_{t+1} = \underset{\theta}{\arg\min} \frac{1}{n}\sum_{i=1}^n \ell(f_\theta(x_i); y_i) + \frac{1}{2\eta_t}D(\theta; \theta_t)$$

for all $t \geq 0$, where $D$ denotes divergence function, and $\theta_0$ is obtained by standard training with SGD. We consider inexact implementation of the proximal step, discussed in (11). Specifically, each proximal step is solved by using SGD, with a batch size of 128, an initial learning rate of 0.1 which is subsequently divided by 5 at the 60th, 120th, and 160th epoch. We consider two divergence functions widely used in practice, defined by

$$D_{\mathrm{LS}}(\theta', \theta) = \frac{1}{2n} \sum_{i=1}^{n} \|f_\theta(x_i) - f_{\theta'}(x_i)\|_2^2 \tag{13}$$

(see Tarvainen and Valpola (2017)), and

$$D_{\mathrm{KL}}(\theta, \theta') = \frac{1}{2n} \sum_{i=1}^{n} \mathrm{KL}\left(f_{\theta'}(x_i) \| f_\theta(x_i)\right) \tag{14}$$

(see Furlanello et al. (2018))[3]. For each of the divergence, we run BPPA with 3 proximal steps, with the proximal stepsize $\eta_t = \eta = 0.025$ for $D_{\mathrm{KL}}$, and $\eta_t = \eta = 0.2$ for $D_{\mathrm{LS}}$ ($\eta_t = 0.025$ gives significantly worse performance). For standard training with SGD, we use a batch size of 128, an initial learning rate of 0.1 further divided by 5 at the 60th, 120th, and 160th epoch. The results are reported in Figure 3.

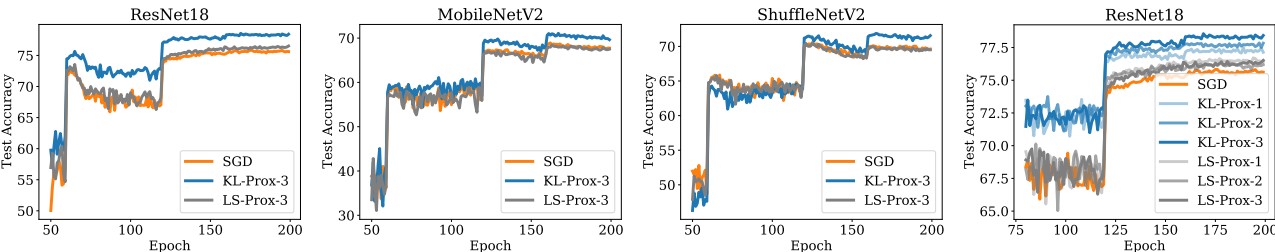

Figure 3: BPPA with divergences $D_{\mathrm{KL}}$ and $D_{\mathrm{LS}}$ on CIFAR-100 dataset. KL-Prox-$k$ denotes learning curve of the $k$-th proximal step with $D_{\mathrm{KL}}$; LS-Prox-$k$ denotes learning curve of the $k$-th proximal step with $D_{\mathrm{LS}}$.

One can clearly see from Figure 3: (1) Across different model architectures, BPPA with $D_{\mathrm{KL}}$ outperforms standard training with SGD; (2) BPPA with $D_{\mathrm{LS}}$ yields negligible differences compared to SGD. The qualitative difference of $D_{\mathrm{KL}}$ and $D_{\mathrm{LS}}$ strongly suggests that the divergence function serves an important role in affecting the model performance learned by BPPA, which we view as an important evidence showing broader applicability of our developed divergence-dependent margin theories. In addition, the learned model with $D_{\mathrm{KL}}$ improves gradually w.r.t the total number of proximal steps. For ResNet-18, the accuracy increases from 75.83% (standard training) to 78.56% after 3 proximal steps – an additional 1.4% improvement over Tf-KD$_{self}$ (see Table 2), which can be viewed as BPPA with one proximal step. We view such findings as the evidence suggesting the scope of algorithmic regularization associated with BPPA goes beyond simple linear models.

| Model | SGD | Tf-KD$_{self}$ |
|---|---|---|
| MobileNetV2 | 68.38 | 70.96 (**+2.58**) |
| ShuffleNetV2 | 70.34 | 72.23 (**+1.89**) |
| ResNet18 | 75.87 | 77.10 (**+1.23**) |
| GoogLeNet | 78.72 | 80.17 (**+1.45**) |
| DenseNet121 | 79.04 | 80.26 (**+1.22**) |

Table 2: Tf-KD$_{self}$ (one-step BPPA) and SGD on CIFAR-100.

At this point it might be worth mentioning a previously proposed method in Yuan et al. (2019), named Teacher-free Knowledge Distillation via self-training (Tf-KD$_{self}$), which is equivalent to BPPA with one

---

[3]It is important to note here that divergences (13) and (14) are not Bregman divergences over the parameter space of $\theta$. Instead, these divergences are induced by Bregman divergences (namely, $\|\cdot - \cdot\|_2^2$ and $\mathrm{KL}(\cdot, \cdot)$, respectively) defined over the prediction space of $f_\theta$. Consequently our discussions for linear model does not directly apply to the nonlinear network considered here.

proximal step, using $D_{\mathrm{KL}}(\theta, \theta')$ as the divergence function. Tf-KD$_{self}$ was shown to improve over SGD for various network architectures on CIFAR-100 and Tiny-ImageNet. We include the reported results on CIFAR-100 therein in Table 2 for completeness.

## 6 Conclusion and Future Direction

To conclude, we have shown that for binary classification task with linearly separable data, the Bregman proximal point algorithm and mirror descent attain a $\|\cdot\|_*$-norm margin that is closely related to the condition number of the distance generating function w.r.t. $\|\cdot\|$-norm. We discuss two directions worthy of future investigations. First, our discussions focus on the Bregman divergences that are defined over the model parameters, while many popular data-dependent divergences are defined over the model output (e.g. prediction confidence). For analyzing the latter class of divergences it seems essential to study the evolution of the trained model in the function space instead of the parameter space. Second, the current analyses focus on linear models, and the extension to nonlinear neural networks requires more delicate definitions of margin and divergence.

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

## A  Discussion on Experiment Section

The observed phenomenon for our second experiment can be explained by Corollary 3.1. Recall that $\mu$ is the maximal $\ell_2$-norm max-margin solution of the mixture of sphere distribution. In Corollary 3.1, take $A = A^{(1)} \coloneqq I_d$, $A = A^{(2)} \coloneqq I_d r^2/d + \mu\mu^\top$ and $A = A^{(3)} \coloneqq \left(I_d r^2/d + \mu\mu^\top\right)^{-1}$ respectively. Note $A^{(1)}, A^{(2)}, A^{(3)}$ by definition are positive definite, hence apply the result of Corollary 3.1, $A^{(3)}$ promotes a solution $\mu^{(3)}$ that has larger directional alignment with $\mu$ (hence achieving better separation), while $A^{(2)}$ promotes a solution $\mu^{(2)}$ that has smaller directional alignment with $\mu$ (hence has worse separation). It remains to note that $\widehat{\Sigma} \to A^{(2)}$, $\widehat{\Sigma}^{-1} \to A^{(3)}$ for $m$ sufficiently large, hence the solutions of BPPA with $D^{(2)}$ and $D^{(3)}$ converge to $\mu^{(2)}$ and $\mu^{(3)}$ for large enough $m$, respectively. In conclusion, BPPA with $D^{(2)}$ converges to a classifier with smaller alignment with $\mu$ (worse separation), while BPPA with $D^{(3)}$ converges to a classifier with larger alignment with $\mu$ (better separation).

## B  Tools from Convex Analysis

We first introduce some useful results in convex analysis, which we use repeatedly in our ensuing developments. Before that, we provide some technical lemmas regarding Bregman divergence and Fenchel conjugate duality.

The first lemma is a folklore result on the duality of Bregman divergence, which shows that the Bregman distance between primal variables $(x, y)$ induced by $w$ is the same as the Bregman distance between the dual variables $(z_y = \nabla w(y), z_x = \nabla w(x))$ induced by $w^*$.

**Lemma B.1** (Duality of Bregman Divergence)**.** *Let* $w : \mathbb{R}^d \to R$ *be strictly convex and differentiable, and* $w^*(\cdot) = \max_x \langle \cdot, x \rangle - w(x)$ *be its convex conjugate. Then we have*

$$D_w(x, y) = D_{w^*}(\nabla w(y), \nabla w(x)), \tag{15}$$

*where* $D_w$ *and* $D_{w^*}$ *denote the Bregman divergence induced by* $w$ *and* $w^*$ *respectively.*

*Proof.* By the definition of conjugate function, we have $w^*(v) = \max_u \langle v, u \rangle - w(u)$. Since $w(\cdot)$ is strictly convex, we have $w^*(\cdot)$ is differentiable, and

$$\nabla w^*(v) = \operatorname*{argmax}_u \langle v, u \rangle - w(u), \Rightarrow \langle v, \nabla w^*(v) \rangle - w(\nabla w^*(v)) = w^*(v), \quad \forall v.$$

Since $w(\cdot)$ is proper, we have $w = (w^*)^*$, which also gives

$$\nabla w(u) = \operatorname*{argmax}_v \langle u, v \rangle - w^*(v), \Rightarrow \langle u, \nabla w(u) \rangle - w^*(\nabla w(u)) = w(u),$$
$$\Rightarrow u = \nabla w^*(\nabla w(u)), \quad \forall u.$$

By the definition of $D_{w^*}$, we have

$$
\begin{aligned}
D_{w^*}&(\nabla w(y), \nabla w(x)) \\
&= w^*(\nabla w(y)) - w^*(\nabla w(x)) - \langle \nabla w^*(\nabla w(x)), \nabla w(y) - \nabla w(x) \rangle \\
&= w^*(\nabla w(y)) - w^*(\nabla w(x)) - \langle x, \nabla w(y) - \nabla w(x) \rangle \\
&= w^*(\nabla w(y)) - \langle \nabla w(y), y \rangle - [w^*(\nabla w(x)) - \langle \nabla w(x), x \rangle] - \langle \nabla w(y), x - y \rangle \\
&= w(x) - w(y) - \langle \nabla w(y), x - y \rangle \\
&= D_w(x, y).
\end{aligned}
$$

$\square$

The second lemma establishes the duality between smoothness and strong convexity w.r.t. to general $\|\cdot\|$-norm. We refer interested readers to Kakade et al. (2009) for the detailed proof.

**Lemma B.2** (Theorem 6, Kakade et al. (2009))**.** *If $f : \mathbb{R}^d \to \mathbb{R}$ is $L$-smooth and $\mu$-strongly convex ($\mu > 0$) with respect to $\|\cdot\|$-norm, then $f^* : \mathbb{R}^d \to \mathbb{R}$ is $\frac{1}{\mu}$-smooth and $\frac{1}{L}$ strongly convex with respect to $\|\cdot\|_*$-norm. Here $f^*(\cdot) = \max_x \langle x, \cdot \rangle - f(x)$ denotes the convex conjugate of $f$, and $(\|\cdot\|, \|\cdot\|_*)$ are a pair of dual norms.*

## C  Proofs in Section 3

*Proof of Theorem 3.2.* For each $m \in \mathbb{Z}^+$, we consider the following simple problem $\mathcal{P}^{(m)}$. First, the data set $\mathcal{S}^{(m)} = \{(x_i, y_i)\}_{i=1}^2 \subset \mathbb{R}^m \times \{+1, -1\}$ contains only two data-points, where $x_1 = z^{(m)}, y_1 = 1$ and $x_2 = -z^{(m)}, y_2 = -1$ for some vector $z^{(m)} \in \mathbb{R}^m$ to be chosen later.

Consider norm $\|\cdot\|^{(m)} = \|\cdot\|_1$, and distance generating function $w^{(m)}(\theta) = \frac{\|\theta\|_2^2}{2}$, both defined on $\mathbb{R}^m$. From the simple identity $\frac{1}{2}\|\theta'\|_2^2 = \frac{1}{2}\|\theta\|_2^2 + \langle \theta, \theta' - \theta \rangle + \frac{1}{2}\|\theta' - \theta\|_2^2$, together with the fact that

$$\|\theta - \theta'\|_2^2 \le \|\theta - \theta'\|_1^2, \quad \|\theta - \theta'\|_2^2 \ge \frac{1}{m}\|\theta - \theta'\|_1^2,$$

we conclude that $w^{(m)}(\cdot)$ is 1-smooth and $\frac{1}{m}$-strongly convex w.r.t. $\|\cdot\|^{(m)}$-norm, and $\sqrt{\mu_{\|\cdot\|}^{(m)}/L_{\|\cdot\|}^{(m)}} = \sqrt{1/m}$. By Corollary 3.1, taking $A = I_m$, we conclude that $\lim_{t \to \infty} \frac{\theta_t}{\|\theta_t\|_2} = u_2^{(m)}$, where

$$u_2^{(m)} = \underset{\|\theta\|_2 \le 1}{\operatorname{argmax}} \min_{(x,y) \in \mathcal{S}^{(m)}} \langle \theta, yx \rangle = \frac{z^{(m)}}{\|z^{(m)}\|_2}$$

is the maximum $\|\cdot\|_2$-norm margin SVM. Hence we have

$$\lim_{t \to \infty} \min_{(x,y) \in \mathcal{S}^{(m)}} \left\langle \frac{\theta_t}{\|\theta_t\|_1}, yx \right\rangle = \left\langle \frac{u_2^{(m)}}{\|u_2^{(m)}\|_1}, z^{(m)} \right\rangle = \frac{\|z^{(m)}\|_2^2}{\|z^{(m)}\|_1}.$$

On the other hand, we can readily verify that $\gamma_{\|\cdot\|_*^{(m)}} := \max_{\|\theta\|^{(m)} \le 1} \langle \theta, z^{(m)} \rangle = \|z^{(m)}\|_\infty$. Thus we conclude that

$$\lim_{t \to \infty} \min_{(x,y) \in \mathcal{S}^{(m)}} \left\langle \frac{\theta_t}{\|\theta_t\|_1}, yx \right\rangle \Big/ \gamma_{\|\cdot\|_*^{(m)}} = \frac{\|z^{(m)}\|_2^2}{\|z^{(m)}\|_1 \|z^{(m)}\|_\infty}.$$

Now taking $z^{(m)} = (1, \frac{1}{\sqrt{m}}, \ldots, \frac{1}{\sqrt{m}})$, one can readily verify that

$$\frac{\|z^{(m)}\|_2^2}{\|z^{(m)}\|_1 \|z^{(m)}\|_\infty} = \frac{2 - 1/m}{\sqrt{m} - 1/\sqrt{m} + 1} \le \frac{2}{\sqrt{m}}, \quad \forall m \ge 1.$$

Thus we conclude our proof with

$$\lim_{t \to \infty} \min_{(x,y) \in \mathcal{S}^{(m)}} \left\langle \frac{\theta_t}{\|\theta_t\|_1}, yx \right\rangle \Big/ \gamma_{\|\cdot\|_*^{(m)}} = \frac{\|z^{(m)}\|_2^2}{\|z^{(m)}\|_1 \|z^{(m)}\|_\infty} \le \frac{2}{\sqrt{m}} = 2\sqrt{\frac{\mu_{\|\cdot\|}^{(m)}}{L_{\|\cdot\|}^{(m)}}}, \quad \forall m \ge 1.$$

$\square$

**Lemma C.1.** *Let $f : \mathbb{R}^d \to \mathbb{R}$ be convex, and $D_w(\theta', \theta) = w(\theta') - w(\theta) - \langle \nabla w(\theta), \theta' - \theta \rangle$ be the Bregman divergence associated with $w$. Letting*

$$\theta_{t+1} \in \underset{\theta \in \mathbb{R}^d}{\operatorname{argmin}} f(\theta) + \frac{1}{2\eta} D_w(\theta, \theta_t), \tag{16}$$

*then for all $\theta \in \mathbb{R}^d$, we have*

$$f(\theta_{t+1}) + \frac{1}{2\eta} D_w(\theta, \theta_{t+1}) \le f(\theta) + \frac{1}{2\eta} D_w(\theta, \theta_t) - \frac{1}{2\eta} D_w(\theta_{t+1}, \theta_t).$$

*Proof.* By the optimality condition of (16), we have

$$\left\langle f'(\theta_{t+1}) + \frac{1}{2\eta}\nabla D_w(\theta_{t+1}, \theta_t), \theta - \theta_{t+1} \right\rangle \geq 0,$$

where $\nabla D_w(\theta_{t+1}, \theta_t)$ denotes the gradient of $D_w(\cdot, \theta_t)$ at $\theta_{t+1}$, and $f'(\theta_{t+1}) \in \partial f(\theta_{t+1})$ denotes the subgradient of $f$ at $\theta_{t+1}$. Now by the definition of the Bregman divergence, we have

$$D_w(\theta, \theta_t) = D_w(\theta_{t+1}, \theta_t) + \langle \nabla D_w(\theta_{t+1}, \theta_t), \theta - \theta_{t+1} \rangle + D_w(\theta, \theta_{t+1}).$$

Combining this with the fact the $\langle f'(\theta_{t+1}), \theta - \theta_{t+1} \rangle \leq f(\theta) - f(\theta_{t+1})$, we conclude our proof. $\square$

**Lemma C.2.** *For proximal point algorithm with Bregman divergence $D_w(\cdot, \cdot)$, where $w(\cdot)$ is $L_{\|\cdot\|}$-smooth w.r.t. to $\|\cdot\|$-norm, we have*

$$\mathcal{L}(\theta_t) - \mathcal{L}(\theta) \leq \frac{L_{\|\cdot\|}}{4\eta t} \|\theta\|^2.$$

*Proof.* Given the update rule

$$\theta_{s+1} \in \underset{\theta}{\arg\min}\, \mathcal{L}(\theta) + \frac{1}{2\eta} D_w(\theta, \theta_s),$$

together with Lemma C.1, wherein we take $f(\cdot) = \mathcal{L}(\cdot)$, and $\theta = \theta_s$, we have that

$$\mathcal{L}(\theta_{s+1}) \leq \mathcal{L}(\theta_s) - \frac{1}{2\eta} D_w(\theta_s, \theta_{s+1}) - \frac{1}{2\eta} D_w(\theta_{s+1}, \theta_s). \tag{17}$$

Since $w(\cdot)$ is convex, we have $D_w(\theta_s, \theta_{s+1}) \geq 0$ and $D_w(\theta_{s+1}, \theta_s) \geq 0$. Thus, we have monotone improvement $\mathcal{L}(\theta_{s+1}) \leq \mathcal{L}(\theta_s)$. On the other hand, by Lemma C.1, we also have

$$\mathcal{L}(\theta_{s+1}) - \mathcal{L}(\theta) \leq \frac{1}{2\eta} D_w(\theta, \theta_s) - \frac{1}{2\eta} D_w(\theta, \theta_{s+1}). \tag{18}$$

Summing up the previous inequality from $s = 0$ to $s = t - 1$, we have

$$\sum_{s=1}^{t} \mathcal{L}(\theta_s) - \mathcal{L}(\theta) \leq \frac{1}{2\eta} D_w(\theta, \theta_0).$$

Additionally, since $w(\cdot)$ is $L_{\|\cdot\|}$-smooth w.r.t. $\|\cdot\|$ norm, we have

$$D_w(\theta, \theta_0) \leq \frac{L_{\|\cdot\|}}{2} \|\theta - \theta_0\|^2.$$

Combining with the fact $\mathcal{L}(\theta_{s+1}) \leq \mathcal{L}(\theta_s)$ for all $s \geq 0$ and $\theta_0 = 0$, we arrive at

$$\mathcal{L}(\theta_t) - \mathcal{L}(\theta) \leq \frac{L_{\|\cdot\|}}{4\eta t} \|\theta\|^2.$$

$\square$

Finally, we show the convergence of loss for binary classification task with separable data. We recall that with separability, the empirical loss $\mathcal{L}$ has infimum zero, but such an infimum *can not be attained.*

♠ *Proof of Theorem 3.1:*

◇ *Proof of Theorem 3.1-(1).* Define $u_R = R\mu_{\|\cdot\|_*}$, where

$$u_{\|\cdot\|_*} = \underset{\|u\| \leq 1}{\arg\max}\, \min_{i \in [n]} \langle u, y_i x_i \rangle,$$

$$\gamma_{\|\cdot\|_*} = \max_{\|u\| \leq 1} \min_{i \in [n]} \langle u, y_i x_i \rangle.$$

By Lemma C.2, we have $\mathcal{L}(\theta_t) \leq \mathcal{L}(u_R) + \frac{L_{\|\cdot\|}}{4\eta t} \|u_R\|^2$. On the other hand, we know that

$$\mathcal{L}(u_R) = \frac{1}{n} \sum_{i=1}^{n} \exp \left\{ -R \left\langle u_{\|\cdot\|_*}, y_i x_i \right\rangle \right\} \leq \exp \left( -R \gamma_{\|\cdot\|_*} \right).$$

Thus, $\mathcal{L}(\theta_t) \leq \exp \left( -R \gamma_{\|\cdot\|_*} \right) + \frac{R^2 L_{\|\cdot\|}}{4\eta t}$. Finally, by taking $R = \frac{\log(t \gamma_{\|\cdot\|_*} \eta)}{\gamma_{\|\cdot\|_*}}$, we have

$$\mathcal{L}(\theta_t) \leq \frac{1}{\gamma_{\|\cdot\|_*} \eta t} + \frac{L_{\|\cdot\|} \log^2 \left( \gamma_{\|\cdot\|_*} \eta t \right)}{4 \gamma_{\|\cdot\|_*}^2 \eta t} = \mathcal{O} \left\{ \frac{L_{\|\cdot\|} \log^2 \left( \gamma_{\|\cdot\|_*} \eta t \right)}{\gamma_{\|\cdot\|_*}^2 \eta t} \right\}. \tag{19}$$

$\square$

We proceed to show the parameter convergence of the proximal point algorithm with constant stepsize.

◇ **Proof of Theorem 3.1-(2).** From (17), we have

$$\sum_{i=1}^{n} \exp \left( - \left\langle \theta_{t+1}, y_i x_i \right\rangle \right) = \mathcal{L}(\theta_{t+1})$$

$$\leq \mathcal{L}(\theta_t) - \frac{1}{2\eta} D_w(\theta_t, \theta_{t+1}) - \frac{1}{2\eta} D_w(\theta_{t+1}, \theta_t)$$

$$= \mathcal{L}(\theta_t) \left\{ 1 - \frac{1}{2\eta \mathcal{L}(\theta_t)} D_w(\theta_t, \theta_{t+1}) - \frac{1}{2\eta \mathcal{L}(\theta_t)} D_w(\theta_{t+1}, \theta_t) \right\}$$

$$\leq \mathcal{L}(\theta_t) \exp \left\{ - \frac{1}{2\eta \mathcal{L}(\theta_t)} D_w(\theta_t, \theta_{t+1}) - \frac{1}{2\eta \mathcal{L}(\theta_t)} D_w(\theta_{t+1}, \theta_t) \right\}.$$

Hence for each $i \in [n]$, the normalized margin can be lower bounded by

$$\left\langle \frac{\theta_{t+1}}{\|\theta_{t+1}\|}, y_i x_i \right\rangle \geq \frac{-\log n - \log \mathcal{L}(\theta_0)}{\|\theta_{t+1}\|} + \frac{1}{2\eta \|\theta_{t+1}\|} \sum_{s=0}^{t} \frac{D_w(\theta_t, \theta_{t+1}) + D_w(\theta_{t+1}, \theta_t)}{\mathcal{L}(\theta_s)}. \tag{20}$$

By optimality condition of the proximal update $\theta_{t+1} \in \arg\min_\theta \mathcal{L}(\theta) + \frac{1}{2\eta} D_w(\theta, \theta_t)$, we have that

$$\frac{1}{2\eta} \left( \nabla w(\theta_{t+1}) - \nabla(\theta_t) \right) + \nabla \mathcal{L}(\theta_{t+1}) = 0, \tag{21}$$

Thus we have

$$\|\nabla w(\theta_{t+1}) - \nabla w(\theta_t)\|_* = \|\nabla w(\theta_{t+1}) - \nabla w(\theta_t)\|_* \|u_{\|\cdot\|_*}\|$$

$$\geq \left\langle \nabla w(\theta_{t+1}) - \nabla w(\theta_t), u_{\|\cdot\|_*} \right\rangle \tag{22}$$

$$= -2\eta \left\langle \nabla \mathcal{L}(\theta_{t+1}), u_{\|\cdot\|_*} \right\rangle = \frac{2\eta}{n} \sum_{i=1}^{n} \left\langle u_{\|\cdot\|_*}, y_i x_i \right\rangle \exp \left( - \left\langle \theta_{t+1}, y_i x_i \right\rangle \right)$$

$$\geq 2\eta \mathcal{L}(\theta_{t+1}) \gamma_{\|\cdot\|_*},$$

where the first inequality follows from $\|u_{\|\cdot\|_*}\| = 1$, the second inequality follows from the Fenchel-Young Inequality, the second equality uses the definition of $\nabla \mathcal{L}(\theta_{t+1})$, and the final inequality follows from $\left\langle u_{\|\cdot\|_*}, y_i x_i \right\rangle \geq \gamma_{\|\cdot\|_*}$.

Now by Lemma B.1, we have $D_w(\theta_t, \theta_{t+1}) = D_{w^*} (\nabla w(\theta_{t+1}), \nabla w(\theta_t))$. Since $w(\cdot)$ is $L_{\|\cdot\|}$-smooth w.r.t. $\|\cdot\|$-norm, by Lemma B.2 we have that $w^*(\cdot)$ is $\frac{1}{L_{\|\cdot\|}}$-strongly convex w.r.t. $\|\cdot\|_*$-norm, which gives us

$$D_{w^*} \left( \nabla w(\theta_{t+1}), \nabla w(\theta_t) \right)$$

$$= w^*(\nabla w(\theta_{t+1})) - w^*(\nabla w(\theta_t)) - \left\langle \nabla w^*(\nabla w(\theta_t)), \nabla w(\theta_{t+1}) - \nabla w(\theta_t) \right\rangle$$

$$\geq \frac{1}{2L_{\|\cdot\|}} \|\nabla w(\theta_{t+1}) - \nabla w(\theta_t)\|_*^2.$$

Thus we have

$$
\begin{aligned}
\sqrt{D_w(\theta_t, \theta_{t+1})} &= \sqrt{D_{w^*}\left(\nabla w(\theta_{t+1}), \nabla w(\theta_t)\right)} \\
&\geq \sqrt{\frac{1}{2L_{\|\cdot\|}}} \|\nabla w(\theta_{t+1}) - \nabla w(\theta_t)\|_* \geq \sqrt{\frac{2}{L_{\|\cdot\|}}} \eta \mathcal{L}(\theta_{t+1}) \gamma_{\|\cdot\|_*},
\end{aligned}
$$

where the last inequality follows from (22).

Together with the fact that $w(\cdot)$ is $\mu_{\|\cdot\|}$-strongly convex w.r.t. $\|\cdot\|$-norm, we have

$$
D_w(\theta_t, \theta_{t+1}) \geq \sqrt{\frac{2}{L_{\|\cdot\|}}} \eta \mathcal{L}(\theta_{t+1}) \gamma_{\|\cdot\|_*} \sqrt{D_w(\theta_t, \theta_{t+1})} \geq \sqrt{\frac{\mu_{\|\cdot\|}}{L_{\|\cdot\|}}} \eta \mathcal{L}(\theta_{t+1}) \gamma_{\|\cdot\|_*} \|\theta_{t+1} - \theta_t\|. \tag{23}
$$

By the same argument, we also have that

$$
D_w(\theta_{t+1}, \theta_t) \geq \sqrt{\frac{\mu_{\|\cdot\|}}{L_{\|\cdot\|}}} \eta \mathcal{L}(\theta_{t+1}) \gamma_{\|\cdot\|_*} \|\theta_{t+1} - \theta_t\|.
$$

Together with (20), we have

$$
\begin{aligned}
&\left\langle \frac{\theta_{t+1}}{\|\theta_{t+1}\|}, y_i x_i \right\rangle \\
&\geq \frac{-\log n - \log \mathcal{L}(\theta_0)}{\|\theta_{t+1}\|} + \frac{1}{2\eta \|\theta_{t+1}\|} \sum_{s=0}^{t} \frac{D_w(\theta_s, \theta_{s+1}) + D_w(\theta_{s+1}, \theta_s)}{\mathcal{L}(\theta_s)} \\
&\geq \frac{-\log n - \log \mathcal{L}(\theta_0)}{\|\theta_{t+1}\|} + \frac{1}{\eta \|\theta_{t+1}\|} \sum_{s=0}^{n} \sqrt{\frac{\mu_{\|\cdot\|}}{L_{\|\cdot\|}}} \eta \frac{\mathcal{L}(\theta_{s+1})}{\mathcal{L}(\theta_s)} \gamma_{\|\cdot\|_*} \|\theta_{s+1} - \theta_s\| \\
&\geq \frac{-\log n - \log \mathcal{L}(\theta_0)}{\|\theta_{t+1}\|} + \frac{1}{\|\theta_{t+1}\|} \sum_{s=0}^{n} \sqrt{\frac{\mu_{\|\cdot\|}}{L_{\|\cdot\|}}} \frac{\mathcal{L}(\theta_{s+1})}{\mathcal{L}(\theta_s)} \gamma_{\|\cdot\|_*} \|\theta_{s+1} - \theta_s\| \\
&\geq \frac{-\log n - \log \mathcal{L}(\theta_0)}{\|\theta_{t+1}\|} + \frac{\gamma_{\|\cdot\|_*}}{\|\theta_{t+1}\|} \sqrt{\frac{\mu_{\|\cdot\|}}{L_{\|\cdot\|}}} \sum_{s=0}^{t} \frac{\mathcal{L}(\theta_{s+1})}{\mathcal{L}(\theta_s)} \|\theta_{s+1} - \theta_s\| \\
&\geq \frac{-\log n - \log \mathcal{L}(\theta_0)}{\|\theta_{t+1}\|} + \gamma_{\|\cdot\|_*} \sqrt{\frac{\mu_{\|\cdot\|}}{L_{\|\cdot\|}}} \left\{ \sum_{s=0}^{t} \frac{\mathcal{L}(\theta_{s+1})}{\mathcal{L}(\theta_s)} \|\theta_{s+1} - \theta_s\| \right\} \Big/ \left\{ \sum_{s=0}^{t} \|\theta_{s+1} - \theta_s\| \right\}.
\end{aligned} \tag{24}
$$

Next, we provide two technical lemmas regarding properties of the iterates produced by proximal point algorithms.

**Lemma C.3.** *The iterates produced by proximal point algorithm $\{\theta_s\}_{s \geq 0}$ satisfy*

$$
\sum_{s=0}^{t} \|\theta_{s+1} - \theta_s\| \geq \|\theta_{t+1}\| \geq \frac{1}{D_{\|\cdot\|_*}} \left\{ \log \left( 2\gamma_{\|\cdot\|_*}^2 \eta t \right) - 2 \log \log \left( \gamma_{\|\cdot\|_*} \eta t \right) \right\}, \tag{25}
$$

$$
1 > \frac{\mathcal{L}(\theta_{s+1})}{\mathcal{L}(\theta_s)} \geq \exp \left\{ -D_{\|\cdot\|_*} \sqrt{\frac{2}{s \mu_{\|\cdot\|} \gamma_{\|\cdot\|_*}} + \frac{L_{\|\cdot\|} \log^2 \left( \gamma_{\|\cdot\|_*} \eta s \right)}{2 \mu_{\|\cdot\|} \gamma_{\|\cdot\|_*}^2 s}} \right\}. \tag{26}
$$

*Proof.* For any $t \geq 0$, since $\theta_0 = 0$, we have

$$
\|\theta_{t+1}\| \leq \sum_{s=0}^{t} \|\theta_{s+1} - \theta_s\|.
$$

Now by the convergence of empirical loss (19), together with the assumption that $\|x_i\|_* \leq D_{\|\cdot\|_*}$,

$$\exp\left(-D\|\theta_{t+1}\|\right) \leq \mathcal{L}(\theta_t) \leq \frac{1}{\gamma_{\|\cdot\|_*}\eta t} + \frac{L_{\|\cdot\|}^2 \log^2\left(\gamma_{\|\cdot\|_*}\eta t\right)}{4\gamma_{\|\cdot\|_*}^2 \eta t} = \mathcal{O}\left\{\frac{L_{\|\cdot\|}\log^2\left(\gamma_{\|\cdot\|_*}\eta t\right)}{4\gamma_{\|\cdot\|_*}^2 \eta t}\right\},$$

Thus we have the following lower bound,

$$\sum_{s=0}^{t}\|\theta_{s+1} - \theta_s\| \geq \|\theta_{t+1}\| \geq \frac{1}{D_{\|\cdot\|_*}}\left\{\log\left(4\gamma_{\|\cdot\|_*}^2 \eta t/L_{\|\cdot\|}\right) - 2\log\log\left(\gamma_{\|\cdot\|_*}\eta t\right)\right\}.$$

On the other hand, we have

$$\frac{\mathcal{L}(\theta_{s+1})}{\mathcal{L}(\theta_t)} = \frac{\sum_{i=1}^{n}\exp\left(-\langle\theta_s, y_i x_i\rangle\right)\exp\left(-\langle\theta_{s+1} - \theta_s, y_i x_i\rangle\right)}{\sum_{i=1}^{n}\exp\left(-\langle\theta_s, y_i x_i\rangle\right)} \geq \exp\left(-\|\theta_{s+1} - \theta_s\|D_{\|\cdot\|_*}\right).$$

We then establish an upper bound on $\|\theta_{s+1} - \theta_s\|$. By (17) and the fact the $\mathcal{L}(\theta) \geq 0$ for all $\theta$, we have

$$D_w(\theta_s, \theta_{s+1}) + D_w(\theta_{s+1}, \theta_s) \leq 2\eta\left(\mathcal{L}(\theta_s) - \mathcal{L}(\theta_{s+1})\right).$$

On the other hand, since $w(\cdot)$ is $\mu_{\|\cdot\|}$-strongly convex, we have

$$\frac{\mu_{\|\cdot\|}}{2}\|\theta_{s+1} - \theta_s\|^2 \leq D_w(\theta_s, \theta_{s+1}), \quad \frac{\mu_{\|\cdot\|}}{2}\|\theta_{s+1} - \theta_s\|^2 \leq D_w(\theta_{s+1}, \theta_s).$$

Thus we obtain

$$\|\theta_{s+1} - \theta_s\| \leq \sqrt{\eta\left(\mathcal{L}(\theta_s) - \mathcal{L}(\theta_{s+1})\right)} \leq \sqrt{\frac{2\eta}{\mu_{\|\cdot\|}}\mathcal{L}(\theta_s)}, \quad \frac{\mathcal{L}(\theta_{s+1})}{\mathcal{L}(\theta_t)} < 1.$$

Together with the convergence of empirical loss in (19), we have

$$\|\theta_{s+1} - \theta_s\| \leq \sqrt{\frac{2}{s\mu_{\|\cdot\|}\gamma_{\|\cdot\|_*}} + \frac{L_{\|\cdot\|}\log^2\left(\gamma_{\|\cdot\|_*}\eta s\right)}{2\mu_{\|\cdot\|}\gamma_{\|\cdot\|_*}^2 s}} = \mathcal{O}\left\{\sqrt{\frac{L_{\|\cdot\|}\log^2\left(\gamma_{\|\cdot\|_*}\eta s\right)}{\mu_{\|\cdot\|}\gamma_{\|\cdot\|_*}^2 s}}\right\}, \tag{27}$$

Thus we have

$$\frac{\mathcal{L}(\theta_{s+1})}{\mathcal{L}(\theta_t)} \geq \exp\left(-\|\theta_{s+1} - \theta_s\|D_{\|\cdot\|_*}\right) = \Omega\left\{\exp\left(-D_{\|\cdot\|_*}\sqrt{\frac{L_{\|\cdot\|}\log^2\left(\gamma_{\|\cdot\|_*}\eta s\right)}{\mu_{\|\cdot\|}\gamma_{\|\cdot\|_*}^2 s}}\right)\right\}.$$

$\square$

Now by (24), we have

$$\left\langle\frac{\theta_{t+1}}{\|\theta_{t+1}\|}, y_i x_i\right\rangle \geq \underbrace{\frac{-\log n - \log\mathcal{L}(\theta_0)}{\|\theta_{t+1}\|}}_{(a)} \tag{28}$$

$$+ \underbrace{\gamma_{\|\cdot\|_*}\sqrt{\frac{\mu_{\|\cdot\|}}{L_{\|\cdot\|}}}\left\{\sum_{s=0}^{t}\frac{\mathcal{L}(\theta_{s+1})}{\mathcal{L}(\theta_s)}\|\theta_{s+1} - \theta_s\|\right\} \Big/ \left\{\sum_{s=0}^{t}\|\theta_{s+1} - \theta_s\|\right\}}_{(b)}.$$

We then derive a large $t$ to ensure $\left\langle\frac{\theta_{t+1}}{\|\theta_{t+1}\|}, y_i x_i\right\rangle \geq (1-\epsilon)\sqrt{\frac{\mu_{\|\cdot\|}}{L_{\|\cdot\|}}}\gamma_{\|\cdot\|_*}$ for all $i \in [n]$.

We first address the term (a) in (28). By (25) in Lemma C.3 we have

$$\|\theta_{t+1}\| \geq \frac{1}{D_{\|\cdot\|_*}}\left\{\log\left(4\gamma_{\|\cdot\|_*}^2 \eta t/L_{\|\cdot\|}\right) - 2\log\log\left(\gamma_{\|\cdot\|_*}\eta t\right)\right\}.$$

Thus combining with $\theta_0 = 0$, (i.e., $\mathcal{L}(\theta_0) = 1$), we need

$$\left| \frac{-\log n - \log \mathcal{L}(\theta_0)}{\|\theta_{t+1}\|} \right| = \frac{\log n}{\|\theta_{t+1}\|} \leq \frac{D_{\|\cdot\|_*} \log n}{\log \left( 4\gamma_{\|\cdot\|_*}^2 \eta t / L_{\|\cdot\|} \right) - 2 \log \log \left( \gamma_{\|\cdot\|_*} \eta t \right)} \tag{29}$$

$$\leq \sqrt{\frac{\mu_{\|\cdot\|}}{L_{\|\cdot\|}}} \frac{\gamma_{\|\cdot\|_*} \epsilon}{2}.$$

One can readily verify that by taking

$$t = \Theta \left\{ \exp \left( \frac{D_{\|\cdot\|_*} \log n}{\epsilon} \gamma_{\|\cdot\|_*} \right) \right\},$$

the previous inequality (29) holds.

It remains to bound term (b) in (28). We need

$$\left\{ \sum_{s=0}^{t} \frac{\mathcal{L}(\theta_{s+1})}{\mathcal{L}(\theta_s)} \|\theta_{s+1} - \theta_s\| \right\} \Bigg/ \left\{ \sum_{s=0}^{t} \|\theta_{s+1} - \theta_s\| \right\} \geq 1 - \frac{\epsilon}{2}. \tag{30}$$

For any $\underline{t}$ such that $0 \leq \underline{t} \leq t - 1$,

$$\left\{ \sum_{s=0}^{t} \frac{\mathcal{L}(\theta_{s+1})}{\mathcal{L}(\theta_s)} \|\theta_{s+1} - \theta_s\| \right\} \Bigg/ \left\{ \sum_{s=0}^{t} \|\theta_{s+1} - \theta_s\| \right\}$$

$$= \frac{1}{\sum_{k=0}^{t} \|\theta_{k+1} - \theta_k\|} \left\{ \sum_{s=0}^{\underline{t}} \|\theta_{s+1} - \theta_s\| \frac{\mathcal{L}(\theta_{s+1})}{\mathcal{L}(\theta_s)} + \sum_{s=\underline{t}+1}^{t} \|\theta_{s+1} - \theta_s\| \frac{\mathcal{L}(\theta_{s+1})}{\mathcal{L}(\theta_s)} \right\}$$

$$> \frac{1}{\sum_{k=0}^{t} \|\theta_{k+1} - \theta_k\|} \left\{ \sum_{s=\underline{t}+1}^{t} \|\theta_{s+1} - \theta_s\| \frac{\mathcal{L}(\theta_{s+1})}{\mathcal{L}(\theta_s)} \right\}.$$

By (26) in Lemma C.3, we have

$$\frac{\mathcal{L}(\theta_{s+1})}{\mathcal{L}(\theta_s)} \geq \exp \left\{ -D_{\|\cdot\|_*} \sqrt{\frac{2}{s\mu_{\|\cdot\|}\gamma_{\|\cdot\|_*}} + \frac{L_{\|\cdot\|} \log^2 \left( \gamma_{\|\cdot\|_*} \eta s \right)}{2\mu_{\|\cdot\|}\gamma_{\|\cdot\|_*}^2 s}} \right\}.$$

One can readily verify that, for $\underline{t} = \widetilde{\Theta} \left( \frac{L_{\|\cdot\|} D_{\|\cdot\|_*}^2}{\mu_{\|\cdot\|}\epsilon^2 \gamma_{\|\cdot\|_*}^2} \right)$, and any $s \geq \underline{t}$

$$\frac{\mathcal{L}(\theta_{s+1})}{\mathcal{L}(\theta_s)} \geq 1 - \frac{\epsilon}{4}. \tag{31}$$

Thus we have

$$\frac{1}{\sum_{k=0}^{t} \|\theta_{k+1} - \theta_k\|} \left\{ \sum_{s=\underline{t}+1}^{t} \|\theta_{s+1} - \theta_s\| \frac{\mathcal{L}(\theta_{s+1})}{\mathcal{L}(\theta_s)} \right\} \geq \left[ 1 - \frac{\sum_{s=0}^{\underline{t}} \|\theta_{s+1} - \theta_s\|}{\sum_{k=0}^{t} \|\theta_{k+1} - \theta_k\|} \right] \left( 1 - \frac{\epsilon}{4} \right). \tag{32}$$

By (27) in the proof of Lemma C.3, we have

$$\|\theta_{s+1} - \theta_s\| \leq \sqrt{\frac{2}{s\mu_{\|\cdot\|}\gamma_{\|\cdot\|_*}} + \frac{L_{\|\cdot\|} \log^2 \left( \gamma_{\|\cdot\|_*} \eta s \right)}{2\mu_{\|\cdot\|}\gamma_{\|\cdot\|_*}^2 s}},$$

Thus we obtain the following upper bound on $\sum_{s=0}^{\underline{t}} \|\theta_{s+1} - \theta_s\|$ that

$$\sum_{s=0}^{\underline{t}} \|\theta_{s+1} - \theta_s\| \leq \sqrt{\frac{2\underline{t}}{\mu_{\|\cdot\|}\gamma_{\|\cdot\|_*}}} + \frac{1}{\gamma_{\|\cdot\|_*}} \sqrt{\frac{L_{\|\cdot\|}}{\mu_{\|\cdot\|}}} \left[ \sqrt{2t_o \log^2(\gamma_{\|\cdot\|_*} \eta)} + \sqrt{\log^2(\underline{t})\underline{t}} \right]$$

$$= \mathcal{O}\left(\frac{\sqrt{t \log^2(t)}}{\gamma_{\|\cdot\|_*}} \sqrt{\frac{L_{\|\cdot\|}}{\mu_{\|\cdot\|}}}\right).$$

Together with (25), we obtain

$$\frac{\sum_{s=0}^{t} \|\theta_{s+1} - \theta_s\|}{\sum_{k=0}^{t} \|\theta_{k+1} - \theta_k\|} \leq D_{\|\cdot\|_*} \frac{\sqrt{\frac{2t}{\mu_{\|\cdot\|}\gamma_{\|\cdot\|_*}}} + \frac{1}{\gamma_{\|\cdot\|_*}}\sqrt{\frac{L_{\|\cdot\|}}{\mu_{\|\cdot\|}}}\left[\sqrt{2t_o \log^2(\gamma_{\|\cdot\|_*}\eta)} + \sqrt{\log^2(t)t}\right]}{\log\left(2\gamma_{\|\cdot\|_*}^2 \eta t\right) - 2\log\log\left(\gamma_{\|\cdot\|_*}\eta t\right)}$$

$$= \mathcal{O}\left(\frac{D_{\|\cdot\|_*}\sqrt{t \log^2(t)}}{\gamma_{\|\cdot\|_*} \log\left(\gamma_{\|\cdot\|_*}^2 \eta t\right)} \sqrt{\frac{L_{\|\cdot\|}}{\mu_{\|\cdot\|}}}\right).$$

Thus, letting $t = \Theta\left\{\exp\left(\frac{D_{\|\cdot\|_*}\sqrt{t}\log t}{\gamma_{\|\cdot\|_*}\epsilon}\sqrt{\frac{L_{\|\cdot\|}}{\mu_{\|\cdot\|}}}\right)/\gamma_{\|\cdot\|_*}^2 \eta\right\}$, we have

$$\frac{\sum_{s=0}^{t} \|\theta_{s+1} - \theta_s\|}{\sum_{k=0}^{t} \|\theta_{k+1} - \theta_k\|} \leq \frac{\epsilon}{4}. \tag{33}$$

In summary, for $\underline{t} = \widetilde{\Theta}\left(\frac{D_{\|\cdot\|_*}^2 L_{\|\cdot\|}}{\epsilon^2 \gamma_{\|\cdot\|_*}^2 \mu_{\|\cdot\|}}\right)$, and

$$t = \max\left\{\underline{t}, \Theta\left\{\exp\left(\frac{D_{\|\cdot\|_*}\sqrt{\underline{t}}\log \underline{t}}{\gamma_{\|\cdot\|_*}\epsilon}\sqrt{\frac{L_{\|\cdot\|}}{\mu_{\|\cdot\|}}}\right)/\gamma_{\|\cdot\|_*}^2 \eta\right\}\right\}$$

$$= \max\left\{\widetilde{\Theta}\left(\frac{D_{\|\cdot\|_*}^2 L_{\|\cdot\|}}{\epsilon^2 \gamma_{\|\cdot\|_*}^2 \mu_{\|\cdot\|}}\right), \Theta\left\{\exp\left(\frac{L_{\|\cdot\|} D_{\|\cdot\|_*}^2}{\mu_{\|\cdot\|}\gamma_{\|\cdot\|_*}^2 \epsilon^2}\log\left(\frac{1}{\epsilon}\right)\right)/\gamma_{\|\cdot\|_*}^2 \eta\right\}\right\},$$

by combining (31), (32), and (33), we have

$$\left\{\sum_{s=0}^{t} \frac{\mathcal{L}(\theta_{s+1})}{\mathcal{L}(\theta_s)} \|\theta_{s+1} - \theta_s\|\right\} \Big/ \left\{\sum_{s=0}^{t} \|\theta_{s+1} - \theta_s\|\right\} \geq (1 - \frac{\epsilon}{4})^2 > 1 - \frac{\epsilon}{2}.$$

Or equivalently, we have (30) holds. Finally, combine (28), (29), and (30), we conclude that there exists

$$t_0 = \max\left\{\widetilde{\mathcal{O}}\left(\frac{L_{\|\cdot\|} D_{\|\cdot\|_*}^2}{\mu_{\|\cdot\|}\epsilon^2\gamma_{\|\cdot\|_*}^2}\right), \mathcal{O}\left\{\exp\left(\frac{L_{\|\cdot\|} D_{\|\cdot\|_*}^2}{\mu_{\|\cdot\|}\gamma_{\|\cdot\|_*}^2 \epsilon^2}\log\left(\frac{1}{\epsilon}\right)\right)/\gamma_{\|\cdot\|_*}^2 \eta\right\}\right\},$$

such that

$$\left\langle \frac{\theta_{t+1}}{\|\theta_{t+1}\|}, y_i x_i \right\rangle \geq (1 - \epsilon)\sqrt{\frac{\mu_{\|\cdot\|}}{L_{\|\cdot\|}}}\gamma_{\|\cdot\|_*}, \quad \forall i \in [n],$$

for all $t \geq t_0$. Hence we conclude our proof. □

□

♠ **Proof of Corollary 3.1.** Define $\phi(u) = \min_{i \in [n]} \langle u, y_i x_i \rangle$. We observe that $\phi(\cdot)$ is concave, and $\phi(\cdot)$ is positive homogenous.

By the previous observations, we claim two properties of $\mathcal{U}^* = \text{argmax}_{\|u\| \leq 1} \phi(u)$:

(a) $\mathcal{U}^*$ is singleton, i.e., $\mathcal{U}^* = \{u_{\|\cdot\|_*}\}$.

(b) $\|u_{\|\cdot\|_*}\| = 1$.

To show (a), suppose that there exits $u_1, u_2 \in \mathcal{U}^*$ and $u_1 \neq u_2$. Let $u_\alpha = \alpha u_1 + (1 - \alpha)u_2$ for any $\alpha \in (0, 1)$. Since $\|\cdot\|$ is a strictly convex function, $\{u : \|u\| \leq 1\}$ is a strictly convex set. Hence $u_\alpha \in \mathrm{int}(\{u : \|u\| \leq 1\})$ and $\|u_\alpha\| < 1$. In addition $\phi(u_\alpha) \geq \alpha\phi(u_1) + (1 - \alpha)\phi(u_2) = \phi(u_1)$ given the fact that $\phi(\cdot)$ is concave and $\phi(u_1) = \phi(u_2)$.

Then we have

$$\phi\left(\frac{u_\alpha}{\|u_\alpha\|}\right) = \frac{1}{\|u_\alpha\|}\phi(u_\alpha) \geq \frac{1}{\|u_\alpha\|}\phi(u_1) > \phi(u_1),$$

which contradicts the fact that $u_1 \in \mathcal{U}^*$. Thus we have (a) holds.

On the other hand, using positive homogeneity, for any $u_* \in \mathcal{U}^*$ such that $\|u_*\| < 1$, we have $\overline{u}_* = u_*/\|u_*\|$ satisfy $\phi(\overline{u}_*) > \phi(u_*)$ and $\|u_*\| = 1$, which again yields a contradiction. Thus we have (b) hold.

To show asymptotic convergence, since we have shown in Theorem 3.1-(2) that $\lim_{t\to\infty} \phi\left(\frac{\theta_t}{\|\theta_t\|}\right) = \max_{\|u\|\leq 1} \phi(u)$, given that $\mathcal{U}^* = \{u_{\|\cdot\|_*}\}$ is a singleton and $\phi(\cdot)$ is continuous, we must have $\lim_{t\to\infty} \frac{\theta_t}{\|\theta_t\|} = u_{\|\cdot\|_*}$. $\qquad \square$

♠ *Proof of Theorem 3.3:*

◇ *Proof of Theorem 3.3-(1).* By (17), we have $\mathcal{L}(\theta_{s+1}) \leq \mathcal{L}(\theta_s) - \frac{1}{2\eta_s}D_w(\theta_s, \theta_{s+1}) - \frac{1}{2\eta_s}D_w(\theta_{s+1}, \theta_s)$. Recall that our stepsize is given by $\eta_s = \frac{\alpha_s}{\mathcal{L}(\theta_s)}$, which gives us

$$\mathcal{L}(\theta_{s+1}) \leq \mathcal{L}(\theta_s) - \frac{1}{2\eta_s}D_w(\theta_s, \theta_{s+1}) - \frac{1}{2\eta_s}D_w(\theta_{s+1}, \theta_s) \tag{34}$$

$$\leq \mathcal{L}(\theta_s)\exp\left\{-\frac{1}{2\eta_s\mathcal{L}(\theta_s)}D_w(\theta_s, \theta_{s+1}) - \frac{1}{2\eta_s\mathcal{L}(\theta_s)}D_w(\theta_{s+1}, \theta_s)\right\}$$

$$= \mathcal{L}(\theta_s)\exp\left\{-\frac{1}{2\alpha_s}D_w(\theta_s, \theta_{s+1}) - \frac{1}{2\alpha_s}D_w(\theta_{s+1}, \theta_s)\right\}. \tag{35}$$

In addition, given the fact that $w(\cdot)$ is $L_{\|\cdot\|}$-smooth w.r.t. $\|\cdot\|$-norm, By Lemma B.2, we have that $w^*(\cdot)$ is $\frac{1}{L_{\|\cdot\|}}$-strongly convex w.r.t. $\|\cdot\|_*$-norm, and hence

$$D_w(\theta_s, \theta_{s+1}) = D_{w^*}(\nabla w(\theta_{s+1}), \nabla w(\theta_s))$$

$$\geq \frac{1}{2L_{\|\cdot\|}}\|\nabla w(\theta_{s+1}) - \nabla w(\theta_s)\|_*^2$$

$$\geq \frac{2\eta_s^2}{L_{\|\cdot\|}}L^2(\theta_{s+1})\gamma_{\|\cdot\|_*}^2 \tag{36}$$

$$= \frac{2\alpha_s^2}{L_{\|\cdot\|}}\left(\frac{\mathcal{L}(\theta_{s+1})}{\mathcal{L}(\theta_s)}\right)^2\gamma_{\|\cdot\|_*}^2,$$

where the first equality holds by Lemma 15, and the second inequality holds by (22). Following the same arguments, we also have $D_w(\theta_{s+1}, \theta_s) \geq \frac{2\alpha_s^2}{L_{\|\cdot\|}}\left(\frac{\mathcal{L}(\theta_{s+1})}{\mathcal{L}(\theta_s)}\right)^2\gamma_{\|\cdot\|_*}^2$. Hence we obtain

$$\frac{\mathcal{L}(\theta_{s+1})}{\mathcal{L}(\theta_s)} \leq \exp\left\{-\left(\frac{\mathcal{L}(\theta_{s+1})}{\mathcal{L}(\theta_s)}\right)^2\frac{2\alpha_s\gamma_{\|\cdot\|_*}^2}{L_{\|\cdot\|}}\right\}.$$

We conclude that $\beta_s := \mathcal{L}(\theta_{s+1})/\mathcal{L}(\theta_s) < 1$. Now for any $\beta \in (0, 1)$, if $\beta_s \geq \beta$, then

$$\beta_s = \frac{\mathcal{L}(\theta_{s+1})}{\mathcal{L}(\theta_s)} \leq \exp\left(-\beta^2\frac{2\alpha_s\gamma_{\|\cdot\|_*}^2}{L_{\|\cdot\|}}\right).$$

Hence we have

$$\frac{\mathcal{L}(\theta_{s+1})}{\mathcal{L}(\theta_s)} \leq \beta_s = \max\left\{\beta, \exp\left(-\beta^2 \frac{2\alpha_s \gamma_{\|\cdot\|_*}^2}{L_{\|\cdot\|}}\right)\right\}, \quad \forall \beta \in (0,1).$$

Since the the previous inequality holds for any $\beta \in (0,1)$, we minimize the right hand side with respect to $\beta$, and obtain

$$\frac{\mathcal{L}(\theta_{s+1})}{\mathcal{L}(\theta_s)} \leq \min_{\beta \in (0,1)} \max\left\{\beta, \exp\left(-\beta^2 \frac{2\alpha_s \gamma_{\|\cdot\|_*}^2}{L_{\|\cdot\|}}\right)\right\} := \beta(\alpha_s). \tag{37}$$

Hence loss $\{\mathcal{L}(\theta_s)\}_{s\geq 0}$ contracts with contraction coefficient $\beta(\alpha_s)$. $\qquad\square$

To proceed, we first establish a lemma stating that $\mathcal{L}(\theta_{s+t})/\mathcal{L}(\theta_s)$ has a lower bound.

**Lemma C.4.** *The iterates produced by Proximal Point Algorithm with varying stepsize $\eta_t = \frac{\alpha_t}{\mathcal{L}(\theta_t)}$ satisfy*

$$\frac{\mathcal{L}(\theta_{t+1})}{\mathcal{L}(\theta_t)} \geq \underline{\beta}(\alpha_t) := \max_{\beta \in (0,1)} \min\left\{\beta, \exp\left(-\alpha_t D_{\|\cdot\|_*}^2 \beta\right)\right\}, \quad \forall t \geq 0. \tag{38}$$

*Proof.* Since $w(\cdot)$ is $\mu_{\|\cdot\|}$-strongly convex w.r.t. $\|\cdot\|$-norm, by Lemma B.2, we have that $w^*(\cdot)$ is $\frac{1}{\mu_{\|\cdot\|}}$-smooth w.r.t. to $\|\cdot\|_*$ norm. Thus,

$$\begin{aligned}
D_w(\theta_s, \theta_{s+1}) &= D_{w^*}(\nabla w(\theta_{s+1}), \nabla w(\theta_s)) \\
&\leq \frac{1}{2\mu_{\|\cdot\|}} \|\nabla w(\theta_{s+1}) - \nabla w(\theta_s)\|_*^2 \\
&= \frac{2\eta_s^2}{\mu_{\|\cdot\|}} \|\nabla \mathcal{L}(\theta_{s+1})\|_*^2 \\
&\leq \frac{2\eta_s^2}{\mu_{\|\cdot\|}} L^2(\theta_{s+1}) D_{\|\cdot\|_*}^2 = \frac{2\alpha_s^2}{\mu_{\|\cdot\|}} \left(\frac{\mathcal{L}(\theta_{s+1})}{\mathcal{L}(\theta_s)}\right)^2 D_{\|\cdot\|_*}^2,
\end{aligned}$$

where the first equality holds by Lemma B.1, the second equality holds by the optimality condition of the proximal update (21), and the last inequality holds by the fact that $\|\nabla \mathcal{L}(\theta_{s+1})\| = \left\|\frac{1}{n}\sum_{i=1}^n \exp\left(-\langle\theta_{s+1}, y_i x_i\rangle\right) y_i x_i\right\| \leq \mathcal{L}(\theta_{s+1}) D_{\|\cdot\|_*}$ and $\|x_i\|_* \leq D_{\|\cdot\|_*}$. Thus by $w(\cdot)$ is $\mu_{\|\cdot\|}$-strongly convex w.r.t. $\|\cdot\|$-norm, i.e., $D_w(\theta_s, \theta_{s+1}) \geq \frac{\mu_{\|\cdot\|}}{2}\|\theta_s - \theta_{s+1}\|^2$, we obtain $\|\theta_{s+1} - \theta_s\| \leq \frac{2\alpha_s}{\mu_{\|\cdot\|}}\frac{\mathcal{L}(\theta_{s+1})}{\mathcal{L}(\theta_s)} D_{\|\cdot\|_*}$.

Thus we have

$$\begin{aligned}
\mathcal{L}(\theta_{s+1}) &= \frac{1}{n}\sum_{i=1}^n \exp\left(-\langle\theta_s, y_i x_i\rangle\right) \exp\left(-\langle\theta_{s+1} - \theta_s, y_i x_i\rangle\right) \\
&\geq \frac{1}{n}\sum_{i=1}^n \exp\left(-\langle\theta_s, y_i x_i\rangle\right) \exp\left(-\|\theta_{s+1} - \theta_s\| D_{\|\cdot\|_*}\right) \\
&\geq \mathcal{L}(\theta_s) \exp\left(-\frac{2\alpha_s D_{\|\cdot\|_*}^2 \mathcal{L}(\theta_{s+1})}{\mu_{\|\cdot\|}\mathcal{L}(\theta_s)}\right).
\end{aligned}$$

Letting $\beta_s := \frac{\mathcal{L}(\theta_{s+1})}{\mathcal{L}(\theta_s)}$, then for any $\beta \in (0,1)$, if $\beta_s \geq \beta$, we have $\frac{\mathcal{L}(\theta_{s+1})}{\mathcal{L}(\theta_s)} \geq \exp\left(-\frac{2\alpha_s D_{\|\cdot\|_*}^2}{\mu_{\|\cdot\|}}\beta\right)$. Thus we conclude that

$$\frac{\mathcal{L}(\theta_{s+1})}{\mathcal{L}(\theta_s)} = \beta_s \geq \min\left\{\beta, \exp\left(-\frac{2\alpha_s D_{\|\cdot\|_*}^2}{\mu_{\|\cdot\|}}\beta\right)\right\}, \quad \forall \beta \in (0,1).$$

Maximizing the right hand side w.r.t. $\beta$, we have

$$\frac{\mathcal{L}(\theta_{s+1})}{\mathcal{L}(\theta_s)} \geq \underline{\beta}(\alpha_s) := \max_{\beta \in (0,1)} \min \left\{ \beta, \exp \left( -\frac{2\alpha_s D_{\|\cdot\|_*}^2}{\mu_{\|\cdot\|}} \beta \right) \right\}. \tag{39}$$

$\square$

◇ **Proof of Theorem 3.3-(2).** We start by (35) and obtain

$$\mathcal{L}(\theta_t) = \mathcal{L}(\theta_{t-1}) \exp \left\{ -\frac{1}{2\alpha_s} D_w(\theta_{t-1}, \theta_t) - \frac{1}{2\alpha_s} D_w(\theta_t, \theta_{t-1}) \right\}$$

$$\leq \mathcal{L}(\theta_0) \exp \left\{ -\frac{1}{2\alpha_s} \sum_{s=0}^{t-1} D_w(\theta_s, \theta_{s+1}) - \frac{1}{2\alpha_s} \sum_{s=0}^{t-1} D_w(\theta_{s+1}, \theta_s) \right\}.$$

From which we obtain that for any $i \in [n]$,

$$\left\langle \frac{\theta_t}{\|\theta_t\|}, y_i x_i \right\rangle \geq \frac{-\log n - \log \mathcal{L}(\theta_0)}{\|\theta_t\|} + \frac{\sum_{s=0}^{t-1} D_w(\theta_s, \theta_{s+1})}{2\alpha_s \|\theta_t\|} + \frac{\sum_{s=0}^{t-1} D_w(\theta_{s+1}, \theta_s)}{2\alpha_s \|\theta_t\|}$$

$$\geq \frac{\sum_{s=0}^{t-1} D_w(\theta_s, \theta_{s+1})}{2\alpha_s \sum_{s=0}^{t-1} \|\theta_{s+1} - \theta_s\|} + \frac{\sum_{s=0}^{t-1} D_w(\theta_{s+1}, \theta_s)}{2\alpha_s \sum_{s=0}^{t-1} \|\theta_{s+1} - \theta_s\|} - \frac{\log n + \log \mathcal{L}(\theta_0)}{\|\theta_t\|} \tag{40}$$

$$\geq \underbrace{\gamma_{\|\cdot\|_*} \sqrt{\frac{\mu_{\|\cdot\|}}{L_{\|\cdot\|}}} \frac{\sum_{s=0}^{t-1} \frac{\mathcal{L}(\theta_{s+1})}{\mathcal{L}(\theta_s)} \|\theta_{s+1} - \theta_s\|}{\sum_{s=0}^{t-1} \|\theta_{s+1} - \theta_s\|}}_{(a)} - \underbrace{\frac{\log n + \log \mathcal{L}(\theta_0)}{\sum_{s=0}^{t-1} \|\theta_{s+1} - \theta_s\|}}_{(b)}, \tag{41}$$

where the second inequality follows from triangle inequality, and the third inequality follows from (23) where we have

$$D_w(\theta_s, \theta_{s+1}) \geq \sqrt{\frac{\mu_{\|\cdot\|}}{L_{\|\cdot\|}}} \eta_s \mathcal{L}(\theta_{s+1}) \gamma_{\|\cdot\|_*} \|\theta_{s+1} - \theta_s\| = \alpha_s \sqrt{\frac{\mu_{\|\cdot\|}}{L_{\|\cdot\|}}} \frac{\mathcal{L}(\theta_{s+1})}{\mathcal{L}(\theta_s)} \gamma_{\|\cdot\|_*},$$

and similarly $D_w(\theta_{s+1}, \theta_s) \geq \alpha_s \sqrt{\frac{\mu_{\|\cdot\|}}{L_{\|\cdot\|}}} \underline{\beta}(\alpha_s) \gamma_{\|\cdot\|_*} \|\theta_{s+1} - \theta_s\|$. We proceed to bound the two terms in (41) separately.

We first give two technical observations. By (36) we readily obtain the first one that

$$\frac{L_{\|\cdot\|}}{2} \|\theta_{s+1} - \theta_s\|^2 \geq D_w(\theta_s, \theta_{s+1}) \geq \frac{2\alpha_s^2}{L_{\|\cdot\|}} \left( \frac{\mathcal{L}(\theta_{s+1})}{\mathcal{L}(\theta_s)} \right)^2 \gamma_{\|\cdot\|_*}^2,$$

or equivalently, $\|\theta_{s+1} - \theta_s\| \geq \frac{2\alpha_s}{L_{\|\cdot\|}} \frac{\mathcal{L}(\theta_{s+1})}{\mathcal{L}(\theta_s)} \gamma_{\|\cdot\|_*}$.

For the second observation, we start from (34), which leads to

$$\frac{\mathcal{L}(\theta_s)}{2\alpha_s} \mu_{\|\cdot\|} \|\theta_s - \theta_{s+1}\|^2 \leq \frac{1}{2\eta_s} \left( D_w(\theta_s, \theta_{s+1}) + D_w(\theta_{s+1}, \theta_s) \right) \leq \mathcal{L}(\theta_s), \tag{42}$$

where the first inequality follows from the definition of $\eta_s$, together with the fact that $w(\cdot)$ is $\mu_{\|\cdot\|}$-strongly convex w.r.t. $\|\cdot\|$-norm. Hence we have $\|\theta_s - \theta_{s+1}\| \leq \sqrt{\frac{2\alpha_s}{\mu_{\|\cdot\|}}}$.

**Claim**: For any $\epsilon > 0$, there exists $\underline{t}$, such that for all $s \geq \underline{t}$, we have $\frac{\mathcal{L}(\theta_{s+1})}{\mathcal{L}(\theta_s)} \geq 1 - \epsilon$.

Given the pervious claim, for any $t > \underline{t}$, we can lower bound (a)

$$\frac{\sum_{s=0}^{t-1} \frac{\mathcal{L}(\theta_{s+1})}{\mathcal{L}(\theta_s)} \|\theta_{s+1} - \theta_s\|}{\sum_{s=0}^{t-1} \|\theta_{s+1} - \theta_s\|} \geq (1 - \epsilon) \frac{\sum_{s=\underline{t}}^{t-1} \|\theta_{s+1} - \theta_s\|}{\sum_{s=0}^{t-1} \|\theta_{s+1} - \theta_s\|}$$

$$\geq (1-\epsilon)\left(1 - \frac{\sum_{s=0}^{t} \|\theta_{s+1} - \theta_s\|}{\sum_{s=0}^{t-1} \|\theta_{s+1} - \theta_s\|}\right)$$

Now by our two observations, we have

$$\sum_{s=0}^{t-1} \|\theta_{s+1} - \theta_s\| \leq \sqrt{\frac{2}{\mu_{\|\cdot\|}}} \sum_{s=0}^{t-1} \sqrt{\alpha_s},$$

$$\sum_{s=0}^{t-1} \|\theta_{s+1} - \theta_s\| \geq \frac{2\gamma_{\|\cdot\|_*}}{L_{\|\cdot\|}} \sum_{s=0}^{t-1} \alpha_s \frac{\mathcal{L}(\theta_s)}{\mathcal{L}(\theta_s)} \geq \frac{2\gamma_{\|\cdot\|_*}}{L_{\|\cdot\|}} \sum_{s=\underline{t}}^{t-1} \alpha_s \frac{\mathcal{L}(\theta_s)}{\mathcal{L}(\theta_s)} \geq \frac{2\gamma_{\|\cdot\|_*}(1-\epsilon)}{L_{\|\cdot\|}} \sum_{s=\underline{t}}^{t-1} \alpha_s.$$

Thus we obtain

$$\frac{\sum_{s=0}^{t-1} \frac{\mathcal{L}(\theta_{s+1})}{\mathcal{L}(\theta_s)} \|\theta_{s+1} - \theta_s\|}{\sum_{s=0}^{t-1} \|\theta_{s+1} - \theta_s\|} \geq (1-\epsilon)\left(1 - \sqrt{\frac{2}{\mu_{\|\cdot\|}}} \frac{L_{\|\cdot\|}}{2\gamma_{\|\cdot\|_*}(1-\epsilon)} \frac{\sum_{s=0}^{t-1} \sqrt{\alpha_s}}{\sum_{s=0}^{t-1} \alpha_s}\right)$$

$$\geq (1-\epsilon)\left(1 - \sqrt{\frac{2}{\mu_{\|\cdot\|}}} \frac{L_{\|\cdot\|}}{\gamma_{\|\cdot\|_*}} \frac{\sum_{s=0}^{t-1} \sqrt{\alpha_s}}{\sum_{s=0}^{t-1} \alpha_s}\right),$$

where the last inequality follows from $\epsilon < \frac{1}{2}$. Now take $\alpha_s = (s+1)^{-1/2}$, we have

$$\frac{\sum_{s=0}^{t-1} \frac{\mathcal{L}(\theta_{s+1})}{\mathcal{L}(\theta_s)} \|\theta_{s+1} - \theta_s\|}{\sum_{s=0}^{t-1} \|\theta_{s+1} - \theta_s\|} \geq (1-\epsilon)\left(1 - \sqrt{\frac{2}{\mu_{\|\cdot\|}}} \frac{L_{\|\cdot\|}}{\gamma_{\|\cdot\|_*}} \mathcal{O}\left(\frac{t^{3/4}}{\sqrt{t} - \sqrt{\underline{t}}}\right)\right)$$

$$= (1-\epsilon)\left(1 - \sqrt{\frac{2}{\mu_{\|\cdot\|}}} \frac{L_{\|\cdot\|}}{\gamma_{\|\cdot\|_*}} \mathcal{O}\left(\frac{t^{3/8}}{\sqrt{t} - \underline{t}^{1/4}}\right)\right) \qquad (43)$$

$$= (1-\epsilon)\left(1 - \sqrt{\frac{2}{\mu_{\|\cdot\|}}} \frac{L_{\|\cdot\|}}{\gamma_{\|\cdot\|_*}} \mathcal{O}\left(\frac{1}{t^{1/8}}\right)\right),$$

where the last two equalities hold by taking $t = \underline{t}^2$.

It remains to determine the size of $\underline{t}$. Note that for this we need

$$\frac{\mathcal{L}(\theta_{s+1})}{\mathcal{L}(\theta_s)} \geq 1 - \epsilon, \quad \forall s \geq \underline{t}.$$

By Lemma C.4, we have $\frac{\mathcal{L}(\theta_{s+1})}{\mathcal{L}(\theta_s)} \geq \underline{\beta}(\alpha_s) := \max_{\beta \in (0,1)} \min\left\{\beta, \exp\left(-\alpha_s D_{\|\cdot\|_*}^2 \beta\right)\right\}$, To make $\underline{\beta}(\alpha_s) = (1-\epsilon)$, we must have

$$\alpha_s = \frac{-\log(1-\epsilon)}{D_{\|\cdot\|_*}^2 \underline{\beta}(\alpha_s)} = \frac{-\log(1-\epsilon)}{D_{\|\cdot\|_*}^2 (1-\epsilon)}.$$

Thus by choosing $\underline{t} \geq \frac{\log^2(1-\epsilon)}{D_{\|\cdot\|_*}^2 (1-\epsilon)^2} = \Theta\left(\frac{1}{D_{\|\cdot\|_*}^2 \epsilon^2}\right)$, we have $\underline{\beta}(\alpha_{\underline{t}}) \geq 1 - \epsilon$. Since $\underline{\beta}(\cdot)$ is a decreasing function, and $\{\alpha_s\}_{s \geq 0}$ is a decreasing sequence, we have $\underline{\beta}(\alpha_s) \geq 1 - \epsilon$ for all $s \geq \underline{t}$.

Thus in summary, by taking

$$t \geq \max\left\{\underline{t}^2, \left(\frac{L_{\|\cdot\|}}{\gamma_{\|\cdot\|_*}\sqrt{\mu_{\|\cdot\|}}\epsilon}\right)^8\right\} = \Theta\left(\left(\frac{L_{\|\cdot\|}}{\gamma_{\|\cdot\|_*}\sqrt{\mu_{\|\cdot\|}}\epsilon}\right)^8\right),$$

from (43), we have control over term (a) as

$$\frac{\sum_{s=0}^{t-1} \frac{\mathcal{L}(\theta_{s+1})}{\mathcal{L}(\theta_s)} \|\theta_{s+1} - \theta_s\|}{\sum_{s=0}^{t-1} \|\theta_{s+1} - \theta_s\|} \geq (1-\epsilon)^2 \geq 1 - 2\epsilon.$$

To control term (b), recall

$$\sum_{s=0}^{t-1} \|\theta_{s+1} - \theta_s\| \geq \frac{2\gamma_{\|\cdot\|_*}(1-\epsilon)}{L_{\|\cdot\|}} \sum_{s=\underline{t}}^{t-1} \alpha_s \geq \frac{\gamma_{\|\cdot\|_*}}{L_{\|\cdot\|}} \sum_{s=\underline{t}}^{t-1} \alpha_s = \Theta\left(\frac{\gamma_{\|\cdot\|_*}}{L_{\|\cdot\|}}\sqrt{t}\right),$$

where the last inequality follows from $\epsilon < \frac{1}{2}$ and $\underline{t} = \underline{t}^2$. Thus by taking

$$t \geq \max\left\{\underline{t}^2, \frac{\log^2 nL_{\|\cdot\|}^3}{\gamma_{\|\cdot\|_*}^4 \mu_{\|\cdot\|}} \frac{1}{\epsilon^2}\right\} = \Theta\left(\frac{1}{D_{\|\cdot\|_*}^4 \epsilon^4}\right),$$

we have a bound for term (b) that $\frac{\log n + \log \mathcal{L}(\theta_0)}{\sum_{s=0}^{t-1}\|\theta_{s+1}-\theta_s\|} \leq \sqrt{\frac{\mu_{\|\cdot\|}}{L_{\|\cdot\|}}}\gamma_{\|\cdot\|_*}\epsilon$.

Putting everything together, there exists $t_0 = \Theta\left(\left(\frac{L_{\|\cdot\|}}{\gamma_{\|\cdot\|_*}\sqrt{\mu_{\|\cdot\|}}\epsilon}\right)^8\right)$, such that for all $t \geq t_0$, we have

$$\left\langle \frac{\theta_t}{\|\theta_t\|}, y_i x_i \right\rangle \geq (1-3\epsilon)\sqrt{\frac{\mu_{\|\cdot\|}}{L_{\|\cdot\|}}}\gamma_{\|\cdot\|_*}, \quad \forall i \in [n].$$

□

◇ **Proof of Theorem 3.3-(2).** To show loss convergence, by (37), we have

$$\mathcal{L}(\theta_t) \leq \exp\left\{\sum_{s=0}^{t-1} \log \beta(\alpha_s)\right\}.$$

Now by the definition of $\beta(\alpha_s)$ we have

$$\beta(\alpha_s) = \exp\left\{-\beta^2(\alpha_s)\frac{2\alpha_s\gamma_{\|\cdot\|_*}^2}{L_{\|\cdot\|}}\right\} \leq \exp\left(-\underline{\beta}^2(\alpha_s)\frac{2\alpha_s\gamma_{\|\cdot\|_*}^2}{L_{\|\cdot\|}}\right).$$

By choosing $\epsilon = \frac{1}{2}$ in the proof of Theorem 3.3-(2), we have that for $t \geq \underline{t} \geq \frac{4\log^2(2)}{D_{\|\cdot\|_*}}$, $\underline{\beta}(\alpha_t) \geq \frac{1}{2}$. Hence for $t \geq \frac{4\log^2(2)}{D_{\|\cdot\|_*}}$, we have $\beta(\alpha_s) \leq \exp\left(-\frac{\alpha_s\gamma_{\|\cdot\|_*}^2}{2L_{\|\cdot\|}}\right)$. Thus we conclude that

$$\mathcal{L}(\theta_t) \leq \exp\left\{\sum_{s=0}^{t-1} \log \beta(\alpha_s)\right\} \leq \exp\left\{-\sum_{s=\underline{t}}^{t-1} \frac{\alpha_s\gamma_{\|\cdot\|_*}^2}{2L_{\|\cdot\|}}\right\} = \mathcal{O}\left(\exp\left(-\frac{\gamma_{\|\cdot\|_*}^2}{L_{\|\cdot\|}}\sqrt{t}\right)\right).$$

□

□

## D   Proofs in Section 4

*Proof of Proposition 4.1.* The proof is a direct consequence of Corollary 4.1 and the proof of Proposition 3.2.   □

♠ *Proof of Theorem 4.1:*

◇ *Proof of Theorem 4.1-(1):* We first show that for small enough stepsize $\eta_t$, we have monotonic improvement, i.e., $\mathcal{L}(\theta_{t+1}) \leq \mathcal{L}(\theta_t)$. Suppose not, and we have $\mathcal{L}(\theta_{t+1}) > \mathcal{L}(\theta_t)$, then given the second-order taylor expansion, there exists $\widetilde{\theta}_t = \alpha_t\theta_t + (1-\alpha_t)\theta_{t+1}$ for some $\alpha_t \in [0,1]$, such that

$$\mathcal{L}(\theta_{t+1}) \leq \mathcal{L}(\theta_t) + \langle \nabla\mathcal{L}(\theta_t), \theta_{t+1} - \theta_t \rangle + \frac{1}{2}(\theta_{t+1} - \theta_t)^\top \nabla^2\mathcal{L}(\widetilde{\theta}_t)(\theta_{t+1} - \theta_t).$$

By some calculation, we have

$$\nabla^2 \mathcal{L}(\widetilde{\theta}_t) = \sum_{i=1}^n \exp\left(-\left\langle \widetilde{\theta}_t, y_i x_i \right\rangle\right) x_i x_i^\top \preccurlyeq D_{\|\cdot\|_2} \mathcal{L}(\widetilde{\theta}_t) I_d.$$

Since $\mathcal{L}(\cdot)$ is a convex function, by the definition of $\widetilde{\theta}_t$ we have

$$\mathcal{L}(\widetilde{\theta}_t) \le \alpha_t \mathcal{L}(\theta_t) + (1 - \alpha_t)\mathcal{L}(\theta_{t+1}) \le \max\{\mathcal{L}(\theta_t), \mathcal{L}(\theta_{t+1})\} = \mathcal{L}(\theta_{t+1}),$$

where the last inequality holds by our assumption $\mathcal{L}(\theta_{t+1}) > \mathcal{L}(\theta_t)$. Hence we have

$$\mathcal{L}(\theta_{t+1}) \le \mathcal{L}(\theta_t) + \langle \nabla \mathcal{L}(\theta_t), \theta_{t+1} - \theta_t \rangle + \frac{\max\{\mathcal{L}(\theta_{t+1}), \mathcal{L}(\theta_t)\}}{2} D_{\|\cdot\|_2} \|\theta_{t+1} - \theta_t\|_2^2 \qquad (44)$$

$$= \mathcal{L}(\theta_t) + \langle \nabla \mathcal{L}(\theta_t), \theta_{t+1} - \theta_t \rangle + \frac{\mathcal{L}(\theta_{t+1})}{2} D_{\|\cdot\|_2} \|\theta_{t+1} - \theta_t\|_2^2.$$

To proceed, we employ the well known three-point lemma in mirror descent analysis, which states the following.

**Lemma D.1.** *Given Bregman divergence $D_w(\cdot, \cdot)$ and mirror descent update*

$$\theta_{t+1} = \operatorname*{argmin}_\theta \langle \nabla \mathcal{L}(\theta_t), \theta - \theta_t \rangle + \frac{1}{2\eta_t} D_w(\theta, \theta_t),$$

*we have*

$$\langle \nabla \mathcal{L}(\theta_{t+1}), \theta_{t+1} - \theta_t \rangle + \frac{1}{2\eta_t} D_w(\theta_{t+1}, \theta_t) \le \frac{1}{2\eta_t} D_w(\theta, \theta_t) - \frac{1}{2\eta_t} D_w(\theta, \theta_{t+1}), \quad \forall \theta.$$

*Proof.* The proof follows exactly the same lines as in Lemma C.1. $\qquad\square$

Since distance generating function $w(\cdot)$ is $\mu_{\|\cdot\|}$-strongly convex w.r.t. $\|\cdot\|$-norm, by the equivalence of norm, we know that there exists $\mu_2 > 0$, such that $w(\cdot)$ is $\mu_2$-strongly convex w.r.t. $\|\cdot\|_2$-norm. That is $D_w(\theta', \theta) \ge \frac{\mu_2}{2} \|\theta' - \theta\|_2^2$ for any $(\theta', \theta)$. Thus combined with Lemma D.1,

$$\mathcal{L}(\theta_{t+1}) \le \mathcal{L}(\theta_t) + \langle \nabla \mathcal{L}(\theta_t), \theta_{t+1} - \theta_t \rangle + \frac{\mathcal{L}(\theta_{t+1})}{2} D_{\|\cdot\|_2} \|\theta_{t+1} - \theta_t\|_2^2$$

$$\le \mathcal{L}(\theta_t) + \langle \nabla \mathcal{L}(\theta_t), \theta_{t+1} - \theta_t \rangle + \frac{\mathcal{L}(\theta_{t+1})}{\mu_2} D_{\|\cdot\|_2} D_w(\theta_{t+1}, \theta_t)$$

$$\le \mathcal{L}(\theta_t) + \langle \nabla \mathcal{L}(\theta_t), \theta_{t+1} - \theta_t \rangle + \frac{1}{2\eta_t} D_w(\theta_{t+1}, \theta_t) - \left(\frac{1}{2\eta_t} - \frac{\mathcal{L}(\theta_{t+1}) D_{\|\cdot\|_2}}{\mu_2}\right) D_w(\theta_{t+1}, \theta_t)$$

$$\le \mathcal{L}(\theta_t) + \langle \mathcal{L}(\theta_t), \theta - \theta_t \rangle + \frac{1}{2\eta_t} D_w(\theta, \theta_t)$$

$$- \frac{1}{2\eta_t} D_w(\theta, \theta_{t+1}) - \left(\frac{1}{2\eta_t} - \frac{\mathcal{L}(\theta_{t+1}) D_{\|\cdot\|_2}}{\mu_2}\right) D_w(\theta_{t+1}, \theta_t).$$

Taking $\theta = \theta_t$ in the previous inequality, we obtain

$$\mathcal{L}(\theta_{t+1}) \le \mathcal{L}(\theta_t) - \frac{1}{2\eta_t} D_w(\theta_t, \theta_{t+1}) - \left(\frac{1}{2\eta_t} - \frac{\mathcal{L}(\theta_{t+1}) D_{\|\cdot\|_2}}{\mu_2}\right) D_w(\theta_{t+1}, \theta_t).$$

Simple rearrangement gives us

$$\mathcal{L}(\theta_{t+1}) \left(1 - \frac{D_w(\theta_{t+1}, \theta_t) D_{\|\cdot\|_2}}{\mu_2}\right) \le \mathcal{L}(\theta_t) \left(1 - \frac{1}{2\eta_t \mathcal{L}(\theta_t)} D_w(\theta_t, \theta_{t+1}) - \frac{1}{2\eta_t \mathcal{L}(\theta_t)} D_w(\theta_{t+1}, \theta_t)\right).$$

One can readily check that whenever $\frac{D_w(\theta_{t+1}, \theta_t) D_{\|\cdot\|_2}}{\mu_2} \le \frac{1}{2\eta_t \mathcal{L}(\theta_t)} \{D_w(\theta_t, \theta_{t+1}) + D_w(\theta_{t+1}, \theta_t)\}$, we have $\mathcal{L}(\theta_{t+1}) \le \mathcal{L}(\theta_t)$ and hence a contradiction. Equivalently, we need

$$\eta_t \le \frac{\mu_2}{2\mathcal{L}(\theta_t)} \cdot \frac{D_w(\theta_t, \theta_{t+1}) + D_w(\theta_{t+1}, \theta_t)}{D_w(\theta_{t+1}, \theta_t) D_{\|\cdot\|_2}},$$

which can be readily satisfied by $\eta_t \leq \frac{\mu_2 \mu_{\|\cdot\|}}{\mathcal{L}(\theta_t) L_{\|\cdot\|} D_{\|\cdot\|_2}}$, by simply observing that $w(\cdot)$ is $L_{\|\cdot\|}$-smooth and $\mu_{\|\cdot\|}$-strongly convex w.r.t. $\|\cdot\|$-norm, which gives us $D_w(\theta_t, \theta_{t+1}) \leq \frac{L_{\|\cdot\|}}{2} \|\theta_{t+1} - \theta_t\|^2$, and $\frac{\mu_{\|\cdot\|}}{2} \|\theta_{t+1} - \theta_t\|^2 \leq D_w(\theta_{t+1}, \theta_t) \leq \frac{L_{\|\cdot\|}}{2} \|\theta_{t+1} - \theta_t\|^2$.

We have shown that $\mathcal{L}(\theta_{t+1}) \leq \mathcal{L}(\theta_t)$ whenever $\eta_t \leq \frac{\mu_2 \mu_{\|\cdot\|}}{\mathcal{L}(\theta_t) L_{\|\cdot\|} D_{\|\cdot\|_2}}$. In addition, by (44), whenever $\eta_t = \eta \leq \frac{\mu_2}{2\mathcal{L}(\theta_t) D_{\|\cdot\|_2}}$, we have for any $\theta$,

$$
\begin{aligned}
\mathcal{L}(\theta_{t+1}) &\leq \mathcal{L}(\theta_t) + \langle \nabla \mathcal{L}(\theta_t), \theta_{t+1} - \theta_t \rangle + \frac{\mathcal{L}(\theta_t)}{2} D_{\|\cdot\|_2} \|\theta_{t+1} - \theta_t\|_2^2 \\
&\leq \mathcal{L}(\theta_t) + \langle \nabla \mathcal{L}(\theta_t), \theta_{t+1} - \theta_t \rangle + \frac{\mathcal{L}(\theta_t)}{\mu_2} D_{\|\cdot\|_2} D_w(\theta_{t+1}, \theta_t) \\
&\leq \mathcal{L}(\theta_t) + \langle \nabla \mathcal{L}(\theta_t), \theta_{t+1} - \theta_t \rangle + \frac{1}{2\eta} D_w(\theta_{t+1}, \theta_t) \\
&\leq \mathcal{L}(\theta_t) + \langle \nabla \mathcal{L}(\theta_t), \theta - \theta_t \rangle + \frac{1}{2\eta} D_w(\theta, \theta_t) - \frac{1}{2\eta} D_w(\theta, \theta_{t+1}) \\
&\leq \mathcal{L}(\theta) + \frac{1}{2\eta} D_w(\theta, \theta_t) - \frac{1}{2\eta} D_w(\theta, \theta_{t+1}).
\end{aligned}
\tag{45}
$$

Thus $\mathcal{L}(\theta_{t+1}) - \mathcal{L}(\theta) \leq \frac{1}{2\eta} D_w(\theta, \theta_t) - \frac{1}{2\eta} D_w(\theta, \theta_{t+1})$, which is the same as the recursion (18) obtained in Lemma C.2. Finally, observe that whenever $\eta_t = \eta \leq \frac{\mu_2 \mu_{\|\cdot\|}}{2\mathcal{L}(\theta_0) L_{\|\cdot\|} D_{\|\cdot\|_2}} = \frac{\mu_2 \mu_{\|\cdot\|}}{2 L_{\|\cdot\|} D_{\|\cdot\|_2}}$, we always have $\mathcal{L}(\theta_{t+1}) \leq \mathcal{L}(\theta_t) \leq \mathcal{L}(\theta_0)$ for all $t \geq 0$. Thus by simple induction argument, $\eta_t = \eta \leq \frac{\mu_2 \mu_{\|\cdot\|}}{2\mathcal{L}(\theta_t) L_{\|\cdot\|} D_{\|\cdot\|_2}}$ for all $t \geq 0$ and all the previous stated statements holds for all $t \geq 0$.

Follow the exact same lines as in Lemma C.2 starting from recursion (18), and the proof of Theorem 3.1-(1), we obtain

$$
\mathcal{L}(\theta_t) \leq \frac{1}{\gamma_{\|\cdot\|_*} \eta t} + \frac{L_{\|\cdot\|} \log^2 (\gamma_{\|\cdot\|_*} \eta t)}{4 \gamma_{\|\cdot\|_*}^2 \eta t} = \mathcal{O} \left\{ \frac{L_{\|\cdot\|} \log^2 (\gamma_{\|\cdot\|_*} \eta t)}{\gamma_{\|\cdot\|_*}^2 \eta t} \right\}.
\tag{46}
$$

$\square$

$\diamond$ *Proof of Theorem 4.1-(2):* Starting from (44), whenever $\eta_t = \eta \leq \frac{\mu_2 \mu_{\|\cdot\|}}{2 L_{\|\cdot\|} D_{\|\cdot\|_2}}$, we have $\mathcal{L}(\theta_{t+1}) \leq \mathcal{L}(\theta_t)$ for all $t \geq 0$ given previous proof of Theorem 4.1-(1), and hence for any $\theta$,

$$
\begin{aligned}
\mathcal{L}(\theta_{t+1}) &\leq \mathcal{L}(\theta_t) + \langle \nabla \mathcal{L}(\theta_t), \theta_{t+1} - \theta_t \rangle + \frac{\mathcal{L}(\theta_t)}{2} D_{\|\cdot\|_2} \|\theta_{t+1} - \theta_t\|_2^2 \\
&\leq \mathcal{L}(\theta_t) + \langle \nabla \mathcal{L}(\theta_t), \theta_{t+1} - \theta_t \rangle + \frac{\mathcal{L}(\theta_t)}{\mu_2} D_{\|\cdot\|_2} D_w(\theta_{t+1}, \theta_t) \\
&= \mathcal{L}(\theta_t) + \langle \nabla \mathcal{L}(\theta_t), \theta - \theta_t \rangle + \frac{1}{2\eta} D_w(\theta_{t+1}, \theta_t) - \left( \frac{1}{2\eta} - \frac{\mathcal{L}(\theta_t)}{\mu_2} D_{\|\cdot\|_2} \right) D_w(\theta_{t+1}, \theta_t) \\
&\leq \mathcal{L}(\theta_t) + \langle \nabla \mathcal{L}(\theta_t), \theta - \theta_t \rangle + \frac{1}{2\eta} D_w(\theta, \theta_t) - \frac{1}{2\eta} D_w(\theta, \theta_{t+1}) \\
&\quad - \left( \frac{1}{2\eta} - \frac{\mathcal{L}(\theta_t)}{\mu_2} D_{\|\cdot\|_2} \right) D_w(\theta_{t+1}, \theta_t),
\end{aligned}
$$

where the last inequality follows from Lemma D.1. Taking $\theta = \theta_t$, we have

$$
\mathcal{L}(\theta_{t+1}) \leq \mathcal{L}(\theta_t) - \frac{1}{2\eta} D_w(\theta_t, \theta_{t+1}) - \left( \frac{1}{2\eta} - \frac{\mathcal{L}(\theta_t)}{\mu_2} D_{\|\cdot\|_2} \right) D_w(\theta_{t+1}, \theta_t)
\tag{47}
$$

$$
\leq \mathcal{L}(\theta_t) \exp \left\{ -\frac{1}{2\eta \mathcal{L}(\theta_t)} D_w(\theta_t, \theta_{t+1}) - \frac{1}{\mathcal{L}(\theta_t)} \left( \frac{1}{2\eta} - \frac{\mathcal{L}(\theta_t)}{\mu_2} D_{\|\cdot\|_2} \right) D_w(\theta_{t+1}, \theta_t) \right\}
$$

$$
\leq \mathcal{L}(\theta_0) \exp \left\{ -\sum_{s=0}^{t} \frac{1}{2\eta \mathcal{L}(\theta_s)} D_w(\theta_s, \theta_{s+1}) - \sum_{s=0}^{t} \frac{1}{\mathcal{L}(\theta_s)} \left( \frac{1}{2\eta} - \frac{\mathcal{L}(\theta_s)}{\mu_2} D_{\|\cdot\|_2} \right) D_w(\theta_{s+1}, \theta_s) \right\}
\tag{48}
$$

Now we can lower bound $D_w(\theta_s, \theta_{s+1})$ by

$$
\begin{aligned}
D_w(\theta_s, \theta_{s+1}) &= \sqrt{D_w(\theta_s, \theta_{s+1})}\sqrt{D_w(\theta_s, \theta_{s+1})} \\
&= \sqrt{D_{w^*}(\nabla w(\theta_{s+1}), \nabla w(\theta_s))}\sqrt{D_w(\theta_s, \theta_{s+1})} \\
&\geq \sqrt{\frac{1}{2L_{\|\cdot\|}}}\|\nabla w(\theta_{s+1}) - \nabla w(\theta_s)\|_*\sqrt{\frac{\mu_{\|\cdot\|}}{2}}\|\theta_{s+1} - \theta_s\|,
\end{aligned}
\tag{49}
$$

where the second equality holds by Lemma B.1, and the final inequality holds by Lemma B.2, together with Condition 1. Note that the optimality condition of mirror descent update gives us

$$
\nabla\mathcal{L}(\theta_s) + \frac{1}{2\eta}\left(\nabla w(\theta_{s+1}) - \nabla w(\theta_s)\right) = 0.
$$

By the definition of $(u_{\|\cdot\|_*}, \gamma_{\|\cdot\|_*})$, we have

$$
\begin{aligned}
\|\nabla w(\theta_{s+1}) - \nabla w(\theta_s)\|_* &= \|\nabla w(\theta_{s+1}) - \nabla w(\theta_s)\|_*\|u_{\|\cdot\|_*}\| \\
&\geq \langle\nabla w(\theta_{s+1}) - \nabla w(\theta_s), u_{\|\cdot\|_*}\rangle \\
&= 2\eta\langle-\nabla\mathcal{L}(\theta_s), u_{\|\cdot\|_*}\rangle \\
&= \eta\frac{2}{n}\sum_{i=1}^n\exp\left(-\langle\theta_s, y_ix_i\rangle\right)\langle y_ix_i, u_{\|\cdot\|_*}\rangle \\
&\geq 2\eta\mathcal{L}(\theta_s)\gamma_{\|\cdot\|_*}.
\end{aligned}
\tag{50}
$$

Thus combining (49) and (50), we conclude that

$$
D_w(\theta_s, \theta_{s+1}) \geq \eta\sqrt{\frac{\mu_{\|\cdot\|}}{L_{\|\cdot\|}}}\mathcal{L}(\theta_s)\gamma_{\|\cdot\|_*}\|\theta_{s+1} - \theta_s\|.
\tag{51}
$$

Follow the exact same lines of argument, we can also show

$$
D_w(\theta_{s+1}, \theta_s) \geq \eta\sqrt{\frac{\mu_{\|\cdot\|}}{L_{\|\cdot\|}}}\mathcal{L}(\theta_s)\gamma_{\|\cdot\|_*}\|\theta_{s+1} - \theta_s\|.
\tag{52}
$$

Using definition $\mathcal{L}(\theta_{t+1}) = \frac{1}{n}\sum_{i=1}^n\exp\left(-\langle\theta_{t+1}, y_ix_i\rangle\right)$, together with (48), (51) and (52), we obtain

$$
\begin{aligned}
&\frac{1}{n}\sum_{i=1}^n\exp\left(-\langle\theta_{t+1}, y_ix_i\rangle\right) \\
&\leq \mathcal{L}(\theta_0)\exp\left\{-\sum_{s=0}^t\frac{1}{2\eta\mathcal{L}(\theta_s)}D_w(\theta_s, \theta_{s+1}) - \sum_{s=0}^t\frac{1}{\mathcal{L}(\theta_s)}\left(\frac{1}{2\eta} - \frac{\mathcal{L}(\theta_s)}{\mu_2}D_{\|\cdot\|_2}\right)D_w(\theta_{s+1}, \theta_s)\right\} \\
&\leq \mathcal{L}(\theta_0)\left\{-\sum_{s=0}^t\frac{1}{2}\sqrt{\frac{\mu_{\|\cdot\|}}{L_{\|\cdot\|}}}\|\theta_{s+1} - \theta_s\|\gamma_{\|\cdot\|_*} - \sum_{s=0}^t\left(\frac{1}{2} - \frac{\mathcal{L}(\theta_s)}{\mu_2}\eta D_{\|\cdot\|_2}\right)\sqrt{\frac{\mu_{\|\cdot\|}}{L_{\|\cdot\|}}}\gamma_{\|\cdot\|_*}\|\theta_{s+1} - \theta_s\|\right\}.
\end{aligned}
$$

Taking logarithm and dividing by $\|\theta_{t+1}\|$ on both sides, we have for any $i \in [n]$,

$$
\begin{aligned}
&\left\langle\frac{\theta_{t+1}}{\|\theta_{t+1}\|}, y_ix_i\right\rangle \\
&\geq \frac{-\log n - \log\mathcal{L}(\theta_0)}{\|\theta_{t+1}\|} + \frac{1}{2}\sqrt{\frac{\mu_{\|\cdot\|}}{L_{\|\cdot\|}}}\frac{\sum_{s=0}^t\|\theta_{s+1} - \theta_s\|\gamma_{\|\cdot\|_*}}{\|\theta_{t+1}\|} \\
&\quad + \sum_{s=0}^t\left(\frac{1}{2} - \frac{\mathcal{L}(\theta_s)}{\mu_2}\eta D_{\|\cdot\|_2}\right)\sqrt{\frac{\mu_{\|\cdot\|}}{L_{\|\cdot\|}}}\gamma_{\|\cdot\|_*}\|\theta_{s+1} - \theta_s\|
\end{aligned}
$$

$$\geq \underbrace{\frac{-\log n - \log \mathcal{L}(\theta_0)}{\|\theta_{t+1}\|}}_{(a)} + \underbrace{\frac{1}{2}\sqrt{\frac{\mu_{\|\cdot\|}}{L_{\|\cdot\|}}} \frac{\sum_{s=0}^{t} \|\theta_{s+1} - \theta_s\| \, \gamma_{\|\cdot\|_*}}{\sum_{s=0}^{t} \|\theta_{s+1} - \theta_s\|}}_{(b)}$$

$$+ \underbrace{\sum_{s=0}^{t} \left( \frac{1}{2} - \frac{\mathcal{L}(\theta_s)}{\mu_2} \eta D_{\|\cdot\|_2} \right) \sqrt{\frac{\mu_{\|\cdot\|}}{L_{\|\cdot\|}}} \gamma_{\|\cdot\|_*} \|\theta_{s+1} - \theta_s\| \Big/ \sum_{s=0}^{t} \|\theta_{s+1} - \theta_s\|}_{(c)}. \tag{53}$$

To get asymptotic convergence, observe that by (46), $\mathcal{L}(\theta_t) \to 0$. Thus we have

$$\|\theta_{t+1}\| \to \infty, \quad \sum_{s=0}^{t} \|\theta_{s+1} - \theta_s\| \geq \|\theta_{t+1}\| \to \infty, \quad \frac{1}{2} - \frac{\mathcal{L}(\theta_s)}{\mu_2} \eta D_{\|\cdot\|_2} \to \frac{1}{2},$$

from which we conclude

$$(a) \to 0, \quad (b) = \frac{1}{2}\sqrt{\frac{\mu_{\|\cdot\|}}{L_{\|\cdot\|}}} \gamma_{\|\cdot\|_*}, \quad (c) \to \frac{1}{2}\sqrt{\frac{\mu_{\|\cdot\|}}{L_{\|\cdot\|}}} \gamma_{\|\cdot\|_*},$$

where we use the simple fact that $\lim_{t\to\infty} \frac{\sum_{s=0}^{t} a_s b_s}{\sum_{s=0}^{t} b_s} = a$ if $\lim_{t\to\infty} \sum_{s=0}^{t} b_s = \infty$ and $\lim_{t\to\infty} a_s = a$. Thus we obtain

$$\lim_{t\to\infty} \left\langle \frac{\theta_{t+1}}{\|\theta_{t+1}\|}, y_i x_i \right\rangle \geq \sqrt{\frac{\mu_{\|\cdot\|}}{L_{\|\cdot\|}}} \gamma_{\|\cdot\|_*}.$$

To get detailed convergence rate, we proceed to treat term (a) and (c) in (53) separately. In order to make $\left\langle \frac{\theta_{t+1}}{\|\theta_{t+1}\|}, y_i x_i \right\rangle \geq (1-\epsilon)\sqrt{\frac{\mu_{\|\cdot\|}}{L_{\|\cdot\|}}} \gamma_{\|\cdot\|_*}$, it suffices to make

$$(a) \leq \sqrt{\frac{\mu_{\|\cdot\|}}{L_{\|\cdot\|}}} \frac{\epsilon \gamma_{\|\cdot\|_*}}{2}, \quad (c) \geq \left(1 - \frac{\epsilon}{2}\right)\sqrt{\frac{\mu_{\|\cdot\|}}{L_{\|\cdot\|}}} \gamma_{\|\cdot\|_*}. \tag{54}$$

To bound term (a), observe that from the loss convergence (46), together with assumption that $\|x_i\|_* \leq D_{\|\cdot\|_*}$,

$$\exp\left(-D\|\theta_{t+1}\|\right) \leq \mathcal{L}(\theta_t) \leq \frac{1}{\gamma_{\|\cdot\|_*} \eta t} + \frac{L_{\|\cdot\|}^2 \log^2\left(\gamma_{\|\cdot\|_*} \eta t\right)}{4\gamma_{\|\cdot\|_*}^2 \eta t} = \mathcal{O}\left\{ \frac{L_{\|\cdot\|} \log^2\left(\gamma_{\|\cdot\|_*} \eta t\right)}{4\gamma_{\|\cdot\|_*}^2 \eta t} \right\}.$$

Thus we have the following lower bound,

$$\sum_{s=0}^{t} \|\theta_{s+1} - \theta_s\| \geq \|\theta_{t+1}\| \geq \frac{1}{D_{\|\cdot\|_*}} \left\{ \log\left(4\gamma_{\|\cdot\|_*}^2 \eta t / L_{\|\cdot\|}\right) - 2\log\log\left(\gamma_{\|\cdot\|_*} \eta t\right) \right\}. \tag{55}$$

One can readily verify that by taking

$$t = \Theta\left\{ \exp\left( \frac{D_{\|\cdot\|_*} \log n}{\epsilon} \gamma_{\|\cdot\|_*} \right) \right\},$$

we have $(a) \leq \sqrt{\frac{\mu_{\|\cdot\|}}{L_{\|\cdot\|}}} \frac{\epsilon \gamma_{\|\cdot\|_*}}{2}$.

To bound (c), it suffices to make

$$\frac{\eta D_{\|\cdot\|_2}}{\mu_2} \frac{\sum_{s=0}^{t} \|\theta_{s+1} - \theta_s\| \mathcal{L}(\theta_s)}{\sum_{s=0}^{t} \|\theta_{s+1} - \theta_s\|} \leq \frac{\epsilon}{2}.$$

Note that for any $0 \leq \underline{t} \leq t-1$, we have

$$\frac{\sum_{s=0}^{t} \|\theta_{s+1} - \theta_s\| \mathcal{L}(\theta_s)}{\sum_{s=0}^{t} \|\theta_{s+1} - \theta_s\|} \leq \frac{\sum_{s=0}^{\underline{t}} \|\theta_{s+1} - \theta_s\| \mathcal{L}(\theta_s)}{\sum_{s=0}^{t} \|\theta_{s+1} - \theta_s\|} + \frac{\sum_{s=\underline{t}+1}^{t} \|\theta_{s+1} - \theta_s\| \mathcal{L}(\theta_s)}{\sum_{s=0}^{t} \|\theta_{s+1} - \theta_s\|}$$

$$\leq \frac{\sum_{s=0}^{\underline{t}} \|\theta_{s+1} - \theta_s\|}{\sum_{s=0}^{t} \|\theta_{s+1} - \theta_s\|} + \frac{\sum_{s=\underline{t}+1}^{t} \|\theta_{s+1} - \theta_s\| \mathcal{L}(\theta_{\underline{t}})}{\sum_{s=0}^{t} \|\theta_{s+1} - \theta_s\|}$$

$$\leq \frac{\sum_{s=0}^{\underline{t}} \|\theta_{s+1} - \theta_s\|}{\sum_{s=0}^{t} \|\theta_{s+1} - \theta_s\|} + \mathcal{L}(\theta_{\underline{t}}),$$

where we use the fact the loss in monotonically improving, thus $\mathcal{L}(\theta_s) \leq \mathcal{L}(\theta_0) = 1$ for all $s \geq 0$, and $\mathcal{L}(\theta_s) \leq \mathcal{L}(\theta_{\underline{t}})$ for all $s \geq \underline{t}$.

On the other hand, taking $\theta = \theta_t$ in (45), we have $\mathcal{L}(\theta_{t+1}) \leq \mathcal{L}(\theta_t) - \frac{1}{2\eta} D_w(\theta_t, \theta_{t+1})$. Since $w(\cdot)$ is $\mu_{\|\cdot\|}$-strongly convex w.r.t. $\|\cdot\|$-norm, and $\mathcal{L}(\theta_t) \geq 0$, we obtain

$$\|\theta_{t+1} - \theta_t\| \leq \sqrt{\frac{2}{\mu_{\|\cdot\|}} (\mathcal{L}(\theta_t) - \mathcal{L}(\theta_{t+1}))} \leq \sqrt{\frac{2}{\mu_{\|\cdot\|}} \mathcal{L}(\theta_t)}.$$

Combining with loss convergence in (46), we have

$$\|\theta_{s+1} - \theta_s\| \leq \sqrt{\frac{2}{s\mu_{\|\cdot\|}\gamma_{\|\cdot\|_*}} + \frac{L_{\|\cdot\|} \log^2\left(\gamma_{\|\cdot\|_*} \eta s\right)}{2\mu_{\|\cdot\|}\gamma_{\|\cdot\|_*}^2 s}},$$

Thus we obtain the following upper bound on $\sum_{s=0}^{t} \|\theta_{s+1} - \theta_s\|$ as

$$\sum_{s=0}^{\underline{t}} \|\theta_{s+1} - \theta_s\| \leq \sqrt{\frac{2\underline{t}}{\mu_{\|\cdot\|}\gamma_{\|\cdot\|_*}}} + \frac{1}{\gamma_{\|\cdot\|_*}} \sqrt{\frac{L_{\|\cdot\|}}{\mu_{\|\cdot\|}}} \left[ \sqrt{t_o 2 \log^2(\gamma_{\|\cdot\|_*}\eta)} + \sqrt{\log^2(\underline{t})\underline{t}} \right]$$

$$= \mathcal{O}\left( \frac{\sqrt{t \log^2(\underline{t})}}{\gamma_{\|\cdot\|_*}} \sqrt{\frac{L_{\|\cdot\|}}{\mu_{\|\cdot\|}}} \right).$$

Together with (55), we obtain

$$\frac{\sum_{s=0}^{\underline{t}} \|\theta_{s+1} - \theta_s\|}{\sum_{k=0}^{t} \|\theta_{k+1} - \theta_k\|} \leq D_{\|\cdot\|_*} \frac{\sqrt{\frac{2\underline{t}}{\mu_{\|\cdot\|}\gamma_{\|\cdot\|_*}}} + \frac{1}{\gamma_{\|\cdot\|_*}} \sqrt{\frac{L_{\|\cdot\|}}{\mu_{\|\cdot\|}}} \left[ \sqrt{t_o 2 \log^2(\gamma_{\|\cdot\|_*}\eta)} + \sqrt{\log^2(\underline{t})\underline{t}} \right]}{\log\left(2\gamma_{\|\cdot\|_*}^2 \eta t\right) - 2 \log\log\left(\gamma_{\|\cdot\|_*}\eta t\right)}$$

$$= \mathcal{O}\left( \frac{D_{\|\cdot\|_*} \sqrt{t \log^2(\underline{t})}}{\gamma_{\|\cdot\|_*} \log\left(\gamma_{\|\cdot\|_*}^2 \eta t\right)} \sqrt{\frac{L_{\|\cdot\|}}{\mu_{\|\cdot\|}}} \right).$$

Thus we can choose $t = \Theta\left\{ \exp\left( \frac{D_{\|\cdot\|_*}\sqrt{\underline{t}}\log \underline{t} D_{\|\cdot\|_2}\eta}{\gamma_{\|\cdot\|_*}\epsilon\mu_2} \sqrt{\frac{L_{\|\cdot\|}}{\mu_{\|\cdot\|}}} \right) \big/ \gamma_{\|\cdot\|_*}^2 \eta \right\}$, so that

$$\frac{\eta D_{\|\cdot\|_2}}{\mu_2} \frac{\sum_{s=0}^{\underline{t}} \|\theta_{s+1} - \theta_s\|}{\sum_{k=0}^{t} \|\theta_{k+1} - \theta_k\|} \leq \frac{\epsilon}{4}. \tag{56}$$

In addition, by the loss convergence (46), we can choose $\underline{t} = \widetilde{\Theta}\left( \frac{L_{\|\cdot\|}D_{\|\cdot\|}}{\mu_2\gamma_{\|\cdot\|_*}^2\epsilon} \right)$, so that $\frac{\eta D_{\|\cdot\|_2}}{\mu_2}\mathcal{L}(\underline{t}) \leq \frac{\epsilon}{4}$. Thus in conclusion, we can choose

$$t = \widetilde{\Theta}\left( \exp\left( \frac{D_{\|\cdot\|_*}^{3/2} D_{\|\cdot\|_2} L_{\|\cdot\|}\eta}{\gamma_{\|\cdot\|_*}^2 \mu_{\|\cdot\|}^{1/2} \mu_2^{3/2} \epsilon^{3/2}} \log\left(\frac{1}{\epsilon}\right) \right) \right),$$

so that $\frac{\eta D_{\|\cdot\|_2}}{\mu_2} \frac{\sum_{s=0}^{\underline{t}} \|\theta_{s+1} - \theta_s\|}{\sum_{k=0}^{t} \|\theta_{k+1} - \theta_k\|} \leq \frac{\epsilon}{4}$.

Finally, we conclude that there exists

$$t_0 = \max \left\{ \Theta \left( \exp \left( \frac{D_{\|\cdot\|_*} \log n}{\epsilon} \gamma_{\|\cdot\|_*} \right) \right), \Theta \left( \exp \left( \frac{D_{\|\cdot\|_*}^{3/2} D_{\|\cdot\|_2} L_{\|\cdot\|} \eta}{\gamma_{\|\cdot\|_*}^2 \mu_{\|\cdot\|}^{1/2} \mu_2^{3/2} \epsilon^{3/2}} \log \left( \frac{1}{\epsilon} \right) \right) \right) \right\},$$

such that for any $t \geq t_0$, we have (54) holds, which implies

$$\left\langle \frac{\theta_{t+1}}{\|\theta_{t+1}\|}, y_i x_i \right\rangle \geq (1 - \epsilon) \sqrt{\frac{\mu_{\|\cdot\|}}{L_{\|\cdot\|}}} \gamma_{\|\cdot\|_*}, \quad \forall i \in [n].$$

$\square$

$\square$

♠ *Proof of Theorem 4.2.*

⋄ *Proof of Theorem 4.2-(1):* Following similar lines as we obtain (53), we can show that for any $i \in [n]$,

$$\left\langle \frac{\theta_{t+1}}{\|\theta_{t+1}\|}, y_i x_i \right\rangle \geq \frac{-\log n - \log \mathcal{L}(\theta_0)}{\|\theta_{t+1}\|} + \frac{1}{2} \sqrt{\frac{\mu_{\|\cdot\|}}{L_{\|\cdot\|}}} \frac{\sum_{s=0}^t \|\theta_{s+1} - \theta_s\| \gamma_{\|\cdot\|_*}}{\|\theta_{t+1}\|}$$

$$+ \sum_{s=0}^t \left( \frac{1}{2} - \frac{\mathcal{L}(\theta_s)}{\mu_2} \eta D_{\|\cdot\|_2} \right) \sqrt{\frac{\mu_{\|\cdot\|}}{L_{\|\cdot\|}}} \gamma_{\|\cdot\|_*} \|\theta_{s+1} - \theta_s\|$$

$$\geq \frac{-\log n - \log \mathcal{L}(\theta_0)}{\|\theta_{t+1}\|} + \frac{1}{2} \sqrt{\frac{\mu_{\|\cdot\|}}{L_{\|\cdot\|}}} \frac{\sum_{s=0}^t \|\theta_{s+1} - \theta_s\| \gamma_{\|\cdot\|_*}}{\sum_{s=0}^t \|\theta_{s+1} - \theta_s\|}$$

$$+ \sum_{s=0}^t \left( \frac{1}{2} - \frac{\mathcal{L}(\theta_s)}{\mu_2} \eta_s D_{\|\cdot\|_2} \right) \sqrt{\frac{\mu_{\|\cdot\|}}{L_{\|\cdot\|}}} \gamma_{\|\cdot\|_*} \|\theta_{s+1} - \theta_s\| \bigg/ \sum_{s=0}^t \|\theta_{s+1} - \theta_s\|$$

$$\geq \sqrt{\frac{\mu_{\|\cdot\|}}{L_{\|\cdot\|}}} \gamma_{\|\cdot\|_*} + \frac{-\log n - \log \mathcal{L}(\theta_0)}{\sum_{s=0}^t \|\theta_{s+1} - \theta_s\|}$$

$$- \sqrt{\frac{\mu_{\|\cdot\|}}{L_{\|\cdot\|}}} \gamma_{\|\cdot\|_*} \sum_{s=0}^t \frac{\mathcal{L}(\theta_s)}{\mu_2} \eta_s D_{\|\cdot\|_2} \|\theta_{s+1} - \theta_s\| \bigg/ \sum_{s=0}^t \|\theta_{s+1} - \theta_s\|. \tag{57}$$

In addition, we have

$$\frac{L_{\|\cdot\|}}{2} \|\theta_{s+1} - \theta_s\|^2 \geq D_w(\theta_s, \theta_{s+1}) = D_{w^*}(\nabla w(\theta_{s+1}), \nabla w(\theta_s))$$

$$\geq \frac{2\eta_s^2}{L_{\|\cdot\|}} L^2(\theta_s) \gamma_{\|\cdot\|_*}^2 = \frac{2\alpha_s^2}{L_{\|\cdot\|}} \gamma_{\|\cdot\|_*}^2, \tag{58}$$

where the first inequality follows from $w(\cdot)$ being $L_{\|\cdot\|}$-smooth w.r.t. $\|\cdot\|$-norm, the second equality follows from Lemma B.1, and the second inequality follows from (50), and the final equality follows from the definition of stepsize $\eta_s = \frac{\alpha_s}{\mathcal{L}(\theta_s)}$. Thus we obtain $\|\theta_{s+1} - \theta_s\| \geq \frac{2\alpha_s \gamma_{\|\cdot\|_*}}{L_{\|\cdot\|}}$. From which we conclude

$$\sum_{s=0}^t \|\theta_{s+1} - \theta_s\| \geq \sum_{s=0}^t \frac{2\gamma_{\|\cdot\|_*}}{L_{\|\cdot\|}} \alpha_s = \Omega \left( \frac{2\gamma_{\|\cdot\|_*}}{L_{\|\cdot\|}} \sqrt{t} \right). \tag{59}$$

On the other hand, whenever $\eta_s \leq \frac{\mu_2 \mu_{\|\cdot\|}}{2\mathcal{L}(\theta_s) L_{\|\cdot\|} D_{\|\cdot\|_2}}$, or equivalently, $\alpha_s \leq \frac{\mu_2 \mu_{\|\cdot\|}}{2L_{\|\cdot\|} D_{\|\cdot\|_2}}$, we have

$$\frac{\mu_{\|\cdot\|}}{2} \|\theta_{s+1} - \theta_s\|^2 \leq D_w(\theta_s, \theta_{s+1}) \leq 2\eta_s \{\mathcal{L}(\theta_s) - \mathcal{L}(\theta_{s+1})\} \leq 2\eta_s \mathcal{L}(\theta_s) = 2\alpha_s,$$

where the first inequality follows from $w(\cdot)$ being $\mu_{\|\cdot\|}$-strongly convex w.r.t. $\|\cdot\|$-norm, the second inequality comes from (47), and the last inequality follows from $\mathcal{L}(\theta_s) \geq 0$ and the definition of $\alpha_s$. Hence we have $\|\theta_s - \theta_{s+1}\| \leq 2\sqrt{\frac{\alpha_s}{\mu_{\|\cdot\|}}}$.

Thus we have

$$
\sum_{s=0}^{t} \frac{\mathcal{L}(\theta_s)}{\mu_2} \eta_s D_{\|\cdot\|_2} \|\theta_{s+1} - \theta_s\| \Big/ \sum_{s=0}^{t} \|\theta_{s+1} - \theta_s\|
$$
$$
\leq \frac{D_{\|\cdot\|_2} L}{\gamma_{\|\cdot\|_*} \mu_2 \sqrt{\mu_{\|\cdot\|}}} \frac{\sum_{s=0}^{t} \alpha_s^{3/2}}{\sum_{s=0}^{t} \alpha_s} = \Omega\left( \frac{D_{\|\cdot\|_2} L}{\gamma_{\|\cdot\|_*} \mu_2 \sqrt{\mu_{\|\cdot\|}}} t^{-1/4} \right). \tag{60}
$$

Thus combining (57), (59), and (60), we conclude that there exists $t_0 = \max\left\{ \Theta\left( \frac{\log nL}{\gamma_{\|\cdot\|_*} \epsilon} \right)^2, \Theta\left( \frac{D_{\|\cdot\|_2} L_{\|\cdot\|}}{\gamma_{\|\cdot\|_*} \mu_2 \sqrt{\mu_{\|\cdot\|}} \epsilon} \right)^4 \right\}$, such that for $t \geq t_0$, we have

$$
\left\langle \frac{\theta_{t+1}}{\|\theta_{t+1}\|}, y_i x_i \right\rangle \geq \sqrt{\frac{\mu_{\|\cdot\|}}{L_{\|\cdot\|}}} \gamma_{\|\cdot\|_*} + \frac{-\log n - \log \mathcal{L}(\theta_0)}{\sum_{s=0}^{t} \|\theta_{s+1} - \theta_s\|}
$$
$$
- \sqrt{\frac{\mu_{\|\cdot\|}}{L_{\|\cdot\|}}} \gamma_{\|\cdot\|_*} \sum_{s=0}^{t} \frac{\mathcal{L}(\theta_s)}{\mu_2} \eta_s D_{\|\cdot\|_2} \|\theta_{s+1} - \theta_s\| \Big/ \sum_{s=0}^{t} \|\theta_{s+1} - \theta_s\|
$$
$$
\geq (1 - \epsilon) \sqrt{\frac{\mu_{\|\cdot\|}}{L_{\|\cdot\|}}} \gamma_{\|\cdot\|_*}, \quad \forall i \in [n].
$$

$\square$

$\diamond$ *Proof of Theorem 4.2-(2):* Follow the exact same line as we show (48), we conclude that with stepsize $\eta_s \leq \frac{\alpha_s}{\mathcal{L}(\theta_s)} \leq \frac{\mu_2}{2 D_{\|\cdot\|_2} \mathcal{L}(\theta_s)}$, we have

$$
\mathcal{L}(\theta_{t+1}) \leq \mathcal{L}(\theta_0) \exp\left\{ -\sum_{s=0}^{t} \frac{1}{2 \eta_s \mathcal{L}(\theta_s)} D_w(\theta_s, \theta_{s+1}) - \sum_{s=0}^{t} \frac{1}{\mathcal{L}(\theta_s)} \left( \frac{1}{2 \eta_s} - \frac{\mathcal{L}(\theta_s)}{\mu_2} D_{\|\cdot\|_2} \right) D_w(\theta_{s+1}, \theta_s) \right\}
$$
$$
\leq \mathcal{L}(\theta_0) \exp\left\{ -\sum_{s=0}^{t} \frac{1}{2 \eta_s \mathcal{L}(\theta_s)} D_w(\theta_s, \theta_{s+1}) \right\} = \mathcal{L}(\theta_0) \exp\left\{ -\sum_{s=0}^{t} \frac{1}{2 \alpha_s} D_w(\theta_s, \theta_{s+1}) \right\}.
$$

In addition, from (58), we have

$$
D_w(\theta_s, \theta_{s+1}) \geq \frac{2 \alpha_s^2}{L_{\|\cdot\|}} \gamma_{\|\cdot\|_*}^2.
$$

Thus we conclude that

$$
\mathcal{L}(\theta_{t+1}) \leq \mathcal{L}(\theta_0) \exp\left\{ -\frac{\gamma_{\|\cdot\|_*}^2}{L_{\|\cdot\|}} \sum_{s=0}^{t} \alpha_s \right\} = \mathcal{O}\left( \exp\left( -\frac{\gamma_{\|\cdot\|_*}^2}{L_{\|\cdot\|}} \sqrt{t} \right) \right).
$$

$\square$

$\square$

