# OpenReview forum: "Implicit Regularization of Bregman Proximal Point Algorithm and Mirror Descent on Separable Data"
_TMLR — Rejected by TMLR_

### Review · Reviewer_uhGq · 2023-09-29

**Summary Of Contributions:**

The paper studies the implicit bias of BPPA algorithm on separable data. Notably, the paper shows that BPPA converges to a margin that depends on the conditional number of the distance generating function. The authors then empirically demonstrated that the algorithms does lead to models have better margin

**Audience:**

Yes

**Broader Impact Concerns:**

It is okay

**Claims And Evidence:**

Yes

**Requested Changes:**

There are two points I would like to see the authors address

1. A main use / implication of the theory is that choosing the right divergence can lead to qualitatively better results, as shown in figure 2 and section 5.3, but exactly how the theory can be used is both unclear and underdiscussed. I want to see a clear derivation of an optimal divergence using the main theorems, given a data distribution (it is okay to do it in a toy example such as that in Section 5.2)

2. I feel that the authors should discuss more on the limitation of the work. The authors showed that the BPPA algorithm has an implicit bias like that of SGD, and the authors used this to argue that BPPA has all sorts of benefits. Then, it is also natural to think that BPPA should also suffer similar problems as SGD, such as convergence to saddles / local maxima. I would like the authors to discuss these potentially implied disadvantages of the BPPA algorithm to give the readers a full picture. For example, see: https://arxiv.org/abs/2107.11774

**Strengths And Weaknesses:**

Strengths:
1. studying the regularization effect of the BPPA algorithm, which is a novel direction
2. shows that the regularization, namely, the margin of the learned classifier depends tightly on the condition number
3. the technical soundness

Weakness:
1. the link from the theory to implications (such as generalization) should have been discussed in much greater detail to improve the broad interest of the work (see the requested changes part -- I am mainly referring to the example in figure 2 and section 5.3)
2. limitations should be discussed more extensively and comprehensively -- see the requested changes

---

### Review · Reviewer_djxk · 2023-11-06

**Summary Of Contributions:**

The authors analyze the proximal Bregman algorithm and mirror descent, with constant and adaptive step size, in the case of linearly separable data. They provide convergence rates for the loss function, and bounds on the linearly separable hyperplane. In addition, the authors provide experiments on CIFAR 100 to support the claim that Bregman algorithms with KL divergence yield better generalization.

**Audience:**

Yes

**Claims And Evidence:**

Yes

**Requested Changes:**

- Merge the theoretical results for clarity (otherwise it feels like a succession of very similar results)
- Detail more extensively the existing literature on the empirical benefits of Bregman-like algorithms for generalization
- More extensive experiments on more datasets or with more modern architectures, and report the influence of the number of proximal steps in Table 2

**Strengths And Weaknesses:**

Strengths:
The paper is overall well-written and very interesting. I was not aware of such generalization properties of Bregman-like algorithms

Weaknesses:

- "Being motivated in theoretical nature, this point is indeed strongly supported by numerous empirical evidences, where successful applications of BPPA hinge upon a careful design of divergence measure" I was not aware of such an empirical success of Bregman algorithms in machine learning. Maybe the authors could develop more precisely where exactly this type of algorithm is useful, and what exactly are the benefits.


- "It is known that the update of dual variables in BPPA or mirror descent has an interpretation closer to the standard gradient descent (Beck, 2017)" could you give a more precise reference than the full book of Beck?


- Thm 3.1: there are very few assumptions in Thm 3.1. In Thm 3.1, you do not have assumptions on the step-size $\eta$, nor separability assumption. Could you comment on that?

- Equation 7: if I understand correctly, the better the conditioning, the larger the margin, and (hopefully) the better the generalization. Is it linked to standard results from the separable data literature?

- Corollary 3.1 is almost a repetition of Thm 3.1., I would remove it for readability, and simply comment that for the chosen norm $|| \cdot ||_{A}$, then $L_{|| \cdot||}$, $\mu_{|| \cdot||}$

- I did not understand Theorem 3.2. I understand that the idea is to show the bound provided in Thm 3.1 is tight, but I do not understand why a sequence of problems is required

- Could you add more comments on Thm 3.3. As I understand, the adaptive step size is uniformly better than the constant step size in terms of loss decrease, and approaching the norm. With constant stepsize, the loss decreases in $\log^2(t)/t$, and requires $exp(-1/\epsilon)$ steps to be $\epsilon$-close to the margin, whereas the varying stepsize can achieve $\epsilon$-close to the margin in polynomial time, and the loss decrease in $\exp(-\sqrt(t))$. Can you comment on this?

- Thm 4.1 is very similar to Thm 3.1, the only difference is in the $t_0$ constant. I would recommend merging both results.

- I have the same remark for Corollary 4.1 as for Corollary 3.1

- Experiments: how exactly do you count the 3 proximal steps of the Bregman algorithm? in other words, what is an epoch for you, do you count 1 pass over the dataset with 3 proximal steps each time as one epoch? or do you normalize by the number of proximal steps for a fair comparison with SGD?

- I think it would add a lot of strength to the paper if authors could provide more extensive experiments to support the use Bregman algorithm for deep learning, maybe by using more modern architectures (see maybe https://paperswithcode.com/sota/image-classification-on-cifar-100)

- What is the influence of the number of proximal steps in Table 2?


Minor
- The classifier is both referred to as $u$ and $f_u$.

---

### Review · Reviewer_j4u6 · 2023-11-12

**Summary Of Contributions:**

This paper examines the implicit regularization effect of the Bregman proximal point algorithm (BPPA).

The setting concerns binary (linear) classification with separable data. This setting has been studied in the implicit regularization literature, for instance, in the work of Soudry et al. 2018, who show that the gradient descent update converges to the maximum margin solution of the SVM optimization program.

The present paper considers the same setting but instead asks if the max-margin classifier can be obtained based on the BPPA.

The first result shows that under a constant step size, one indeed obtains a lower bound on the margin following the BPPA updates.

The convergence is super polynomial, however, meaning that to attain a $1-\epsilon$ close max-margin solution, the number of iterations needs to scale as $exp(-\epsilon^{-2})$. This result is not improvable, which is demonstrated in a lower bound example next.

The second result shows how to fix this issue by using varying step sizes.

The paper then proceeds to extend the above results to mirror descent.

Finally, experiments are used to illustrate the BBPA updates in a number of empirical settings.

**Audience:**

Yes

**Claims And Evidence:**

Yes

**Requested Changes:**

- A high-level description of the proof sketches would be very helpful. Currently all of the proofs can only be found in the appendix, making it hard to understand the proof details.

- A summary of results (like a table), including a comparison with prior work, would be helpful.

- The paper is missing some related work from the literature. See the following reference and related work therein:

Du, Simon S., Wei Hu, and Jason D. Lee. "Algorithmic regularization in learning deep homogeneous models: Layers are automatically balanced." Advances in neural information processing systems 31 (2018).

**Strengths And Weaknesses:**

Strengths:

S1) A natural extension of a well-known result by Soudry et al. (2018) to the BBPA updates and the mirror descent procedure.

S2) The paper is clearly written.

S3) Detailed proofs are provided in the appendix.

Weaknesses:

W1) The proof details are all included in the appendix, making it difficult to read them in line with the theorems. This might limit the transparency of the paper.

W2) Numerical study seems detached from the theoretical claims in prior sections.

---

### Decision · Action_Editor_b17G · 2024-02-04

**Recommendation:** Reject

**Comment:**

Unfortunately, the authors were not able to submit a revised paper in time. This case was discussed with the editors, and the decision was made to reject with encouragement to resubmit when the authors are ready. Given the detailed feedback from the reviewers, I believe that these issues can easily be cleaned up with a major revision.

**Audience:**

This is well-suited to TMLR.

**Claims And Evidence:**

Some reviewers found the writing to be too brief and hard to follow in some places. For example, references to support some of the claims are missing, and in some places the theorem statements are too vague.

I believe that after a major revision, these issues can easily be cleaned up (the reviewers have offered very specific feedback), however, the authors did not submit a response or revision. I encourage the authors to revise and resubmit their paper at a later time.

**Resubmission Of Major Revision:**

The authors may consider submitting a major revision at a later time.